# GFPT2/GFAT2 and AMDHD2 act in tandem to control the hexosamine pathway

**Virginia Kroef**[1†], **Sabine Ruegenberg**[1,2†], **Moritz Horn**[1,3], **Kira Allmeroth**[1], **Lena Ebert**[4,5,6], **Seyma Bozkus**[2], **Stephan Miethe**[1], **Ulrich Elling**[7], **Bernhard Schermer**[4,5,6], **Ulrich Baumann**[2], **Martin Sebastian Denzel**[1,5,6,8]\*

[1]Max Planck Institute for Biology of Ageing, Cologne, Germany; [2]Institute of Biochemistry, University of Cologne, Cologne, Germany; [3]JLP Health GmbH, Austria and Acus Laboratories GmbH, Cologne, Germany; [4]Department II of Internal Medicine, University of Cologne, Faculty of Medicine and University Hospital Cologne, Cologne, Germany; [5]Center for Molecular Medicine Cologne (CMMC) Faculty of Medicine and University Hospital Cologne, University of Cologne, Cologne, Germany; [6]CECAD - Cluster of Excellence Faculty of Medicine and University Hospital Cologne, University of Cologne, Cologne, Germany; [7]IMBA - Institute of Molecular Biotechnology of the Austrian Academy of Science Vienna Biocenter, Vienna, Austria; [8]Altos Labs, Cambridge, United Kingdom

\*For correspondence:
mdenzel@altoslabs.com

†These authors contributed equally to this work

Competing interest: The authors declare that no competing interests exist.

**Abstract** The hexosamine biosynthetic pathway (HBP) produces the essential metabolite UDP-GlcNAc and plays a key role in metabolism, health, and aging. The HBP is controlled by its rate-limiting enzyme glutamine fructose-6-phosphate amidotransferase (GFPT/GFAT) that is directly inhibited by UDP-GlcNAc in a feedback loop. HBP regulation by GFPT is well studied but other HBP regulators have remained obscure. Elevated UDP-GlcNAc levels counteract the glycosylation toxin tunicamycin (TM), and thus we screened for TM resistance in haploid mouse embryonic stem cells (mESCs) using random chemical mutagenesis to determine alternative HBP regulation. We identified the N-acetylglucosamine deacetylase AMDHD2 that catalyzes a reverse reaction in the HBP and its loss strongly elevated UDP-GlcNAc. To better understand AMDHD2, we solved the crystal structure and found that loss-of-function (LOF) is caused by protein destabilization or interference with its catalytic activity. Finally, we show that mESCs express AMDHD2 together with GFPT2 instead of the more common paralog GFPT1. Compared with GFPT1, GFPT2 had a much lower sensitivity to UDP-GlcNAc inhibition, explaining how AMDHD2 LOF resulted in HBP activation. This HBP configuration in which AMDHD2 serves to balance GFPT2 activity was also observed in other mESCs and, consistently, the GFPT2:GFPT1 ratio decreased with differentiation of human embryonic stem cells. Taken together, our data reveal a critical function of AMDHD2 in limiting UDP-GlcNAc production in cells that use GFPT2 for metabolite entry into the HBP.

## Editor's evaluation

This manuscript describes an interesting regulation of the hexosamine biosynthetic pathway (HBP) that is relative specific to the mouse embryonic stem cells (mESC). HBP produces UDP-N-acetylglucosamine, which is used in various protein glycosylation events, thus regulating many biological pathways. Understanding this pathway and its regulation is thus of fundamental significance.

## Introduction

The hexosamine biosynthetic pathway (HBP) is an anabolic branch of glycolysis consuming about 2%–3% of cellular glucose (*Marshall et al., 1991*; *Ghosh et al., 1960*). It provides substrates for various posttranslational modification (PTM) reactions and has been strongly associated with stress resistance and longevity as well as cell growth and transformation (*Denzel et al., 2014*; *Wellen et al., 2010*; *Yamashita et al., 1985*). Thus, the HBP plays an essential role for metabolic adaptations and cellular homeostasis (*McClain and Crook, 1996*).

In the first and rate limiting step of the HBP, glutamine fructose-6-phosphate amidotransferase (GFPT) converts fructose-6-phosphate (Frc6P) and L-glutamine (L-Gln) to D-glucosamine-6-phosphate (GlcN6P) (*Ghosh et al., 1960*). The two mammalian GFPT paralogs GFPT1 and GFPT2 show 75%–80% amino acid sequence identity (*Oki et al., 1999*). While GFPT1 is ubiquitously expressed, GFPT2 is reported to be predominantly expressed in the nervous system. Notably, GlcN6P can be converted to Frc6P by glucosamine-6-phosphate deaminase 1 and 2 (GNPDA1/2), shunting metabolites back into glycolysis (*Arreola et al., 2003*). In the second step of the HBP, glucosamine-phosphate N-acetyltransferase (GNA1) acetylates GlcN6P to N-acetylglucosamine-6-phosphate (GlcNAc-6P) using acetyl-CoA as the acetyl donor (*Wang et al., 2008*). This reaction is also presumed to be reversible through deacetylation of GlcNAc6P (*Weidanz et al., 1996*; *Bergfeld et al., 2012*). After isomerization into GlcNAc-1-phosphate (GlcNAc-1P) mediated by GlcNAc phosphomutase (PGM3), UTP is used in a final step by UDP-N-acetylglucosamine pyrophosphorylase (UAP1) to synthesize the final product uridine 5′-diphosphate-N-acetyl-D-glucosamine (UDP-GlcNAc) (*Ricciardiello et al., 2018*; *Mio et al., 1998*). UDP-GlcNAc can be reversibly interconverted to its epimer uridine 5′-diphosphate-N-acetyl-D-galactosamine (UDP-GalNAc) by the enzyme UDP-galactose-4′-epimerase (GALE) and the pool of both metabolites is termed UDP-HexNAc (*Thoden et al., 2001*). The HBP is the only source for UDP-GlcNAc and relies on substrates from carbon, nitrogen, fatty-acid, and energy metabolism. It is therefore optimally positioned as a metabolic sensor that can modulate downstream cellular signaling through UDP-GlcNAc dependent PTMs (*Marshall et al., 1991*).

UDP-GlcNAc is a precursor of several important biomolecules, such as chitin, peptidoglycans, and glycosaminoglycans, and for a number of dynamic glycosylation events. Mucin-type O-glycosylation plays an important role in the extracellular matrix (*Hanisch, 2001*). N-linked-glycosylation orchestrates protein folding in the endoplasmic reticulum and is therefore crucial in protein homeostasis (*Parodi, 2000*). N-glycans further contribute to the cell surface glycocalyx as structural components of proteins (*Martinez-Seara Monne et al., 2013*). Finally, the addition of single GlcNAc moieties to Thr/Ser residues, termed O-GlcNAcylation, occurs dynamically on hundreds of proteins, thus modulating a variety of downstream pathways (*Hart, 1997*). Surprisingly, this dynamic PTM is accomplished by a single protein, O-GlcNAc transferase (OGT), and O-GlcNAcase (OGA) is the only known enzyme to remove O-GlcNAc modifications (*Haltiwanger et al., 1992*; *Dong and Hart, 1994*). While it is known that these glycosylation reactions are limited by intracellular UDP-GlcNAc, how the HBP is regulated to adapt UDP-GlcNAc levels according to nutrient availability is poorly understood. Due to the diverse function of UDP-GlcNAc, alterations in its abundance can have detrimental effects resulting in pathological conditions like diabetes, cancer, cardiovascular diseases, and neurodegenerative diseases (*Marshall et al., 1991*; *Oikari et al., 2018*; *Arnold et al., 1996*; *Champattanachai et al., 2007*).

In a previous chemical mutagenesis screen in *Caenorhabditis elegans*, we isolated mutants resistant to the toxin tunicamycin (TM) as a proxy for enhanced protein quality control and found that TM resistant mutants were enriched for longevity (*Denzel et al., 2014*). TM is a competitive inhibitor of UDPGlcNAc:dolichylphosphate GlcNAc-1-phosphotransferase (GPT), which catalyzes the first step of Nglycan synthesis utilizing UDP-GlcNAc (*Heifetz et al., 1979*). TM thus disrupts N-glycosylation and leads to proteins misfolding and proteotoxic stress (*Parodi, 2000*). We found that single amino acid substitutions in GFPT1 result in gain-of-function (GOF) due to loss of UDP-GlcNAc feedback inhibition, elevating cellular UDP-GlcNAc levels and thereby counteracting TM toxicity (*Ruegenberg et al., 2020*). By introducing the same GOF mutation in GFPT1 of mouse neuroblastoma Neuro2a (N2a) cells, we confirmed a conserved mechanism (*Horn et al., 2020*), suggesting that screening for TM resistance might be a suitable unbiased means to analyze the HBP through genetic approaches in mammalian cells. Based on this knowledge, we aimed to identify novel regulators of the HBP in mammalian cells, which could serve as potential drug targets for future therapeutic interventions.

In this study, we combined chemical mutagenesis with whole-exome sequencing in haploid murine cells and identified the N-acetylglucosamine-6-phosphate deacetylase AMDHD2 (Amidohydrolase Domain Containing 2) as a novel regulator of the HBP. Through AMDHD2 deletion, we discovered a configuration of the HBP that uses GFPT2 as the key enzyme. Functionally, GFPT2 shows a lower sensitivity to UDP-GlcNAc feedback inhibition compared to GFPT1 therefore requiring AMDHD2 to balance HBP metabolic flux.

## Results

### Chemical mutagenesis screen for TM resistance in haploid mESCs identifies AMDHD2

Elevated HBP activity and high UDP-GlcNAc concentrations suppress TM toxicity, making TM resistance a proxy for HBP activity in genetic screens. To investigate HBP regulation in mammalian cells, we therefore performed an unbiased TM resistance screen. The mutagen N-ethyl-N-nitrosourea (ENU) induces single-nucleotide variants that enable a screen at amino acid resolution. Thus, we used ENU in haploid cells, which uniquely enable the identification of recessive alleles (*Elling et al., 2011*; *Horn et al., 2018*; *Allmeroth et al., 2021*). In order to reach a high degree of saturation, 27 million AN3-12 mouse embryonic stem cells (mESCs) were used for mutagenesis. This was followed by TM selection using a wild-type (WT) lethal dose (0.5 µg/ml) for 3 weeks (*Figure 1A*). Twenty-nine resistant clones were randomly selected and picked to grow isogenic mutant lines. Whole-exome sequencing was done with four clones, which showed strong TM resistance (*Figure 1—figure supplement 1A*). Two clones revealed independent missense mutations in the *Amdhd2* coding sequence (*Figure 1—figure supplement 1B*). A second round of whole-exome sequencing of the remaining 25 clones revealed in total 11 independent amino acid substitutions at 10 distinct positions in *Amdhd2* (38% of sequenced clones) (*Figure 1B*, *Figure 1—figure supplement 1B*). Surprisingly, we did not identify any mutations in the HBP's rate limiting enzymes *Gfpt1* or *Gfpt2*. In addition, we performed a random insertional mutagenesis screen using an enhanced gene trapping system, which was previously established in haploid mESCs (*Figure 1—figure supplement 1C*; *Elling et al., 2017*). After selection for TM resistance and mapping of the insertion site by Sanger sequencing, we identified *Amdhd2* in 4 of the 20 analyzed clones (*Figure 1C*, *Figure 1—figure supplement 1C*). Since disruption of the *Amdhd2* locus by transgene insertion was sufficient to mediate TM resistance, we concluded that the identified mutations are loss-of-function (LOF) mutations. To corroborate that *Amdhd2* disruption leads to TM resistance, we generated *Amdhd2* KO mutants in diploid WT AN3-12 cells using CRISPR/Cas9. We generated and validated a specific AMDHD2 antibody, which confirmed a successful KO of AMDHD2 (*Figure 1D*). To exclude off target effects, we generated three independent *Amdhd2* KO lines using distinct guide combinations. All homozygous *Amdhd2* KO cells showed significant TM resistance compared to WT cells, confirming AMDHD2 LOF as causal for TM resistance (*Figure 1E and F*, *Figure 1—figure supplement 1D*).

### Disruption of *Amdhd2* mediates TM resistance via elevated HBP flux

AMDHD2 is an amidohydrolase that plays a potential role in the HBP by catalyzing the deacetylation of GlcNAc-6P in the 'reverse' direction of the pathway (*White and Pasternak, 1967*). However, a role of AMDHD2 in modulating cellular UDP-GlcNAc levels has not been recognized before. We hypothesized that AMDHD2 LOF might increase UDP-GlcNAc levels leading to TM resistance (*Figure 2A*). To test this, we measured UDP-GlcNAc levels via ionic chromatography/mass spectrometry (IC-MS) and found that TM resistant mutants identified in the insertional mutagenesis screen (clones 1–4) as well as the CRISPR/Cas9-generated AMDHD2 KO mutants showed a significant increase in UDP-GlcNAc concentrations (*Figure 2B*, *Figure 2—figure supplement 1*). These data indicate that the TM resistance is mediated by elevated HBP product availability due to reduced catabolism of GlcNAc-6P. To further corroborate a causal role of AMDHD2 mutation in elevated UDP-GlcNAc levels and the accompanying TM resistance, we performed rescue experiments with N-terminally FLAG-HA-tagged human AMDHD2 (hAMDHD2). We compared the expression of WT hAMDHD2 and, based on information from the bacterial homolog N-acetylglucosamine-6-phosphate deacetylase (NagA) (*Hall et al., 2007*), a potential catalytically inactive mutant with a D294A substitution (hAMDHD2 D294A) in control WT and AMDHD2 KO mESCs (*Figure 2—figure supplement 2A,B*). Overexpression of WT or mutant

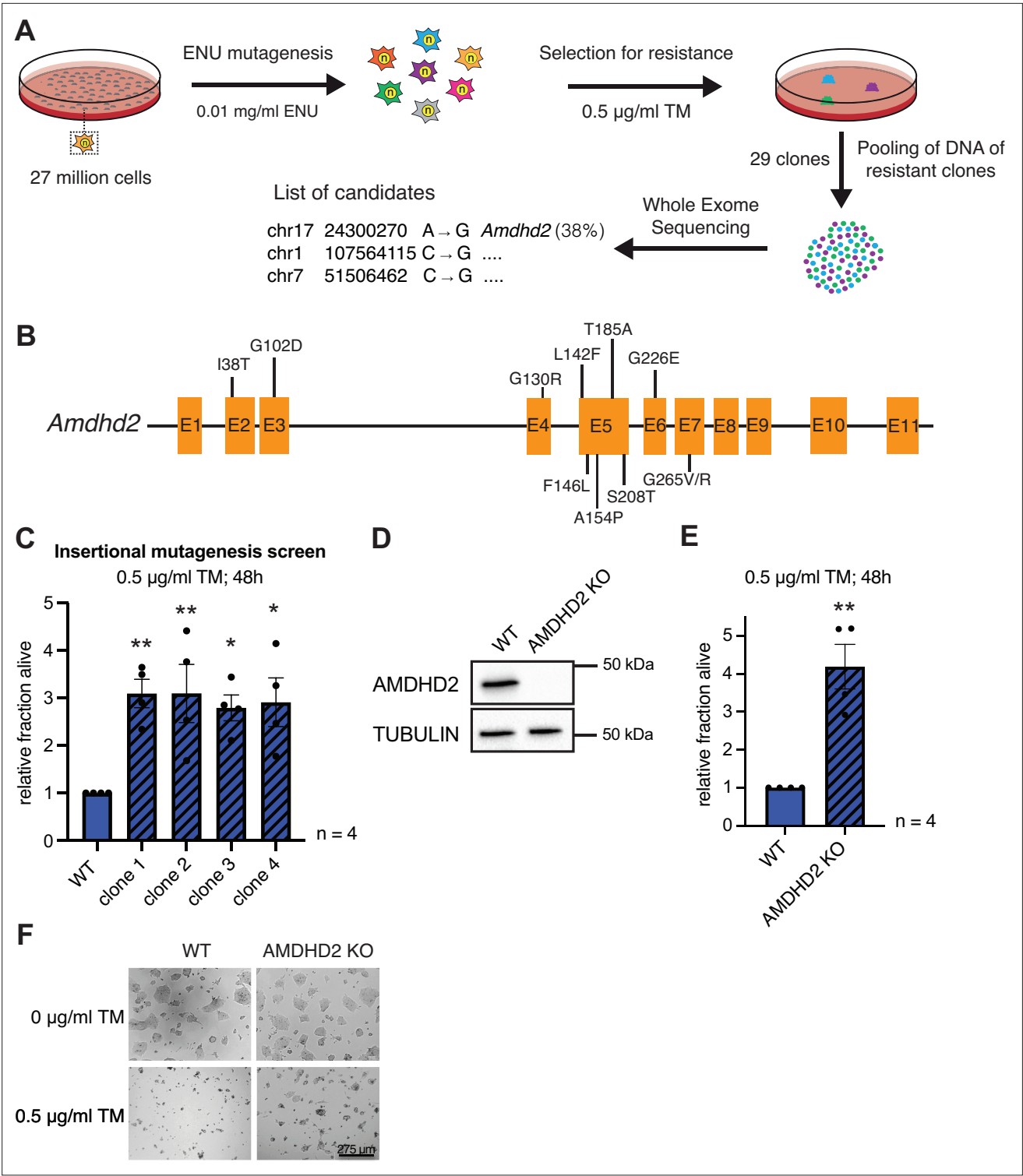

**Figure 1.** Chemical mutagenesis screen for tunicamycin (TM) resistance in haploid mESCs identifies AMDHD2. (**A**) Schematic representation of experimental workflow for TM resistance screen using ENU mutagenesis in combination with whole-exome sequencing. (**B**) Schematic representation of the mouse *Amdhd2* locus. Amino acid substitutions identified in the screen are highlighted. (**C**) Cell viability (XTT assay) of four TM resistant clones (clones 1–4) identified via insertional mutagenesis compared to control wild-type (WT) AN3-12 mESCs. Cells were treated with 0.5 µg/ml TM for 48 hr (mean ± SEM, n=4, *p<0.05, **p<0.01, one-way ANOVA Dunnett's posttest). (**D**) Western blot analysis of CRISPR/Cas9 generated AMDHD2 KO AN3-12 mESCs compared to WT cells. (**E**) Cell viability (XTT assay) of WT and AMDHD2 KO AN3-12 cells treated with 0.5 µg/ml TM for 48 hr (mean ± SEM, n=4,

*Figure 1 continued on next page*

*Figure 1 continued*

**p<0.01, unpaired t-test). (**F**) Representative images of WT and AMDHD2 KO AN3-12 cells treated with 0.5 µg/ml TM for 48 hr or respective control. Scale bar, 275 µm. mESC, mouse embryonic stem cell.

The online version of this article includes the following source data and figure supplement(s) for figure 1:

**Source data 1.** Raw data.

**Figure supplement 1.** Chemical mutagenesis screen for tunicamycin (TM) resistance in haploid mESCs identifies AMDHD2.

**Figure supplement 1—source data 1.** Raw data.

hAMDHD2 did not affect UDP-GlcNAc levels or TM resistance in WT mESCs (*Figure 2C and D*). However, in AMDHD2 KO cells, only re-expression of functional WT hAMDHD2 reduced UDP-GlcNAc levels, while overexpression of the inactive hAMDHD2 D294A mutant still resulted in significantly elevated UDP-GlcNAc levels compared to WT cells. This observation was functionally supported by TM resistance assays using the same cell lines; overexpression of mutant hAMDHD2 D294A in the AMDHD2 KO background had no effect, but expression of WT hAMDHD2 reduced TM resistance. Taken together, these data emphasize the relevance of functional AMDHD2 for HBP activity and they show that AMDHD2 deletion results in TM resistance via increased HBP activity.

To better understand the physiological consequences of HBP activation through AMDHD2 regulation, we disrupted the *Amdhd2* locus to generate a KO mouse (*Figure 2—figure supplement 2A-C*). Although the *Amdhd2* mutation distributed in Mendelian ratios in the offspring, no viable homozygous *Amdhd2* KO pups were weaned (*Figure 2E*), indicating a recessive mutation. Heterozygous animals, however, did not show any macroscopic changes, although further analysis is still missing and alterations on a behavioral, anatomical, histological, or molecular level cannot be excluded. Homozygous *Amdhd2* KO embryos showed early embryonic lethality, indicating an essential function of AMDHD2 during development. Taken together, we identified AMDHD2 as novel regulator of the HBP important in mESCs and for embryonic development.

## Structural and biochemical characterization of human AMDHD2

Until now, no structure of eukaryotic AMDHD2 was available and functional properties of human AMDHD2 remain largely unexplored. Therefore, we performed a structural and a biochemical characterization of human AMDHD2. Initial apo AMDHD2 crystals diffracted poorly and no structure could be solved. Based on homology to bacterial NagA, human AMDHD2 is likely to bind a divalent cation in the active site, potentially stabilizing the protein and supporting co-crystallization. Consequently, we analyzed the stabilizing effect of several divalent cations. Addition of $CoCl_2$, $NiCl_2$, and $ZnCl_2$ to the size-exclusion chromatography (SEC) buffer increased the thermal stability of AMDHD2 by 3–4°C (*Figure 3A*). Moreover, we tested the influence of $CoCl_2$, $NiCl_2$, and $ZnCl_2$ on the deacetylase activity of AMDHD2. For that purpose, the metal co-factor of AMDHD2 was first removed by incubation with EDTA and then $CoCl_2$, $NiCl_2$, or $ZnCl_2$ were added back. Addition of $MgCl_2$ served as negative control, while an untreated AMDHD2 was used as positive control. Both $CoCl_2$ and $ZnCl_2$ restored and $ZnCl_2$ even increased AMDHD2 activity (*Figure 3B*). Thus, $Co^{2+}$ or $Zn^{2+}$ might be the metal co-factor in human AMDHD2. We next tested co-crystallization of AMDHD2 with $ZnCl_2$ or $CoCl_2$. While no crystals formed in the presence of $CoCl_2$, the co-crystallization with $ZnCl_2$ yielded needle clusters in several conditions. Optimized crystals diffracted to a resolution limit of 1.84 Å (AMDHD2+Zn) or 1.90 Å (AMDHD2+Zn+GlcN6P). The data collection and refinement statistics are summarized in *Table 1*. Human AMDHD2 is organized into two domains, a deacetylase domain responsible for the conversion of GlcNAc-6P into GlcN6P and a second small domain with unknown function (DUF) (*Figure 3C*, *Figure 3—figure supplement 1*). Residues from both the N-terminus and the C-terminus contribute to the DUF domain. The structure of AMDHD2 was almost completely modeled into the electron density map except for some N-terminal (1–5) and C-terminal residues (407–409). In the asymmetric unit, AMDHD2 forms a dimer through direct interactions of the deacetylase domains with an interface of 1117 Å$^2$ and this dimeric assembly was judged as biological relevant by the EPPIC server (*Bliven et al., 2018*). Although the dimer is formed by a rather small interface, this conformation is supported by the crystallographic B-factors, which show low values at the interface, indicating a mutual stabilization (*Figure 3—figure supplement 2*) and by dynamic light scattering (DLS) measurements, confirming the presence of AMDHD2 dimers in solution (*Figure 3—figure supplement 3*).

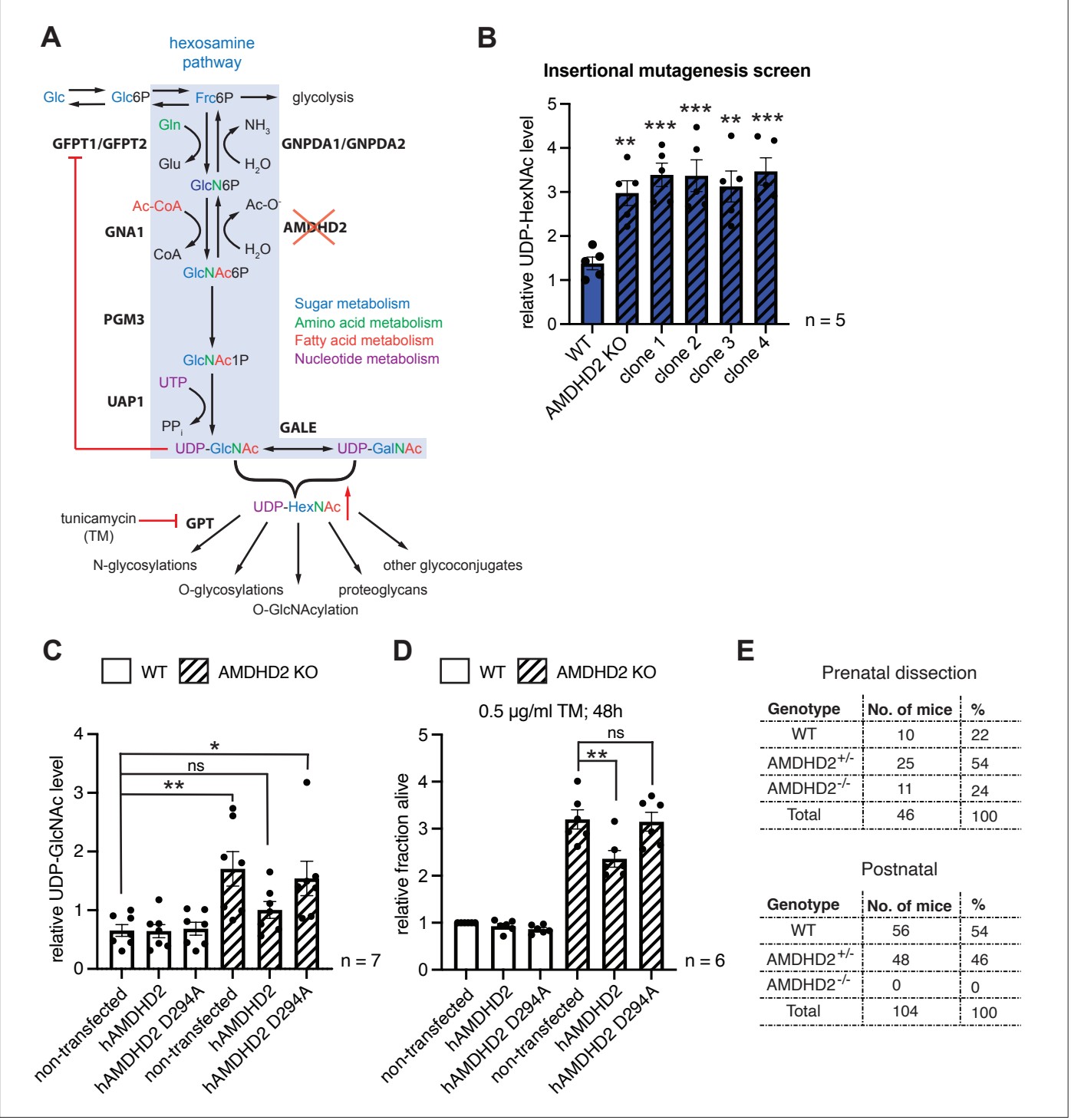

**Figure 2.** Disruption of *Amdhd2* mediates tunicamycin (TM) resistance via elevated HBP flux. (**A**) Schematic overview of the hexosamine pathway (blue box). The intermediate Frc6P from glycolysis is converted to UDP-GlcNAc, which is a precursor for glycosylation reactions. The enzymes are glutamine fructose-6-phosphate amidotransferase (GFPT1/2), glucosamine-6-phosphate N-acetyltransferase (GNA1), phosphoglucomutase (PGM3), UDP-N-acetylglucosamine pyrophosphorylase (UAP1), glucosamine-6-phosphate deaminase (GNPDA1/2), N-acetylglucosamine deacetylase (AMDHD2), UDP-GlcNAc:dolichylphosphate GlcNAc-1-phosphotransferase (GPT), and UDP-galactose-4'-epimerase (GALE). Red line indicates negative feedback inhibition of GFPT by UDP-GlcNAc. UDP-HexNAc is a precursor for various glycosylation reactions including N-glycosylation, O-glycosylation, and O-GlcNAcylation or the synthesis of proteoglycans and other glycoconjugates. N-glycosylation is inhibited by TM. (**B**) ICMS analysis of UDP-HexNAc levels of four TM resistant clones (clones 1–4) generated in insertional mutagenesis screen compared to control WT and AMDHD2 KO AN3-12 mESCs (mean ± SEM, n=5, **p<0.01, ***p<0.001, one-way ANOVA Dunnett's posttest). (**C**) IC-MS analysis of UDP-GlcNAc levels in WT and AMDHD2 KO AN3-12 cells

*Figure 2 continued*

expressing WT FLAGHA-hAMDHD2 (hAMDHD2) and mutant FLAG-HA-hAMDHD2 D294A (hAMDHD2 D294A) (mean ± SEM, n=7, *p<0.05, **p<0.01, ns=not significant, one-way ANOVA Tukey posttest). (**D**) Cell viability (XTT assay) of WT and AMDHD2 KO AN3-12 mESCs expressing WT FLAG-HA-hAMDHD2 (hAMDHD2) and mutant FLAG-HA-hAMDHD2 D294 (hAMDHD2 D294A). Cells were treated with 0.5 μg/ml TM for 48 hr (mean ± SEM, n=6, **p<0.01, ns=not significant, one-way ANOVA Tukey posttest). (**E**) Genotyping results for the AMDHD2 deletion in dissected (E7–8) embryos and weaned mice. HBP, hexosamine biosynthetic pathway; mESC, mouse embryonic stem cell; WT, wild-type.

The online version of this article includes the following source data and figure supplement(s) for figure 2:

**Source data 1.** Raw data.

**Figure supplement 1.** Disruption of *Amdhd2* mediates tunicamycin resistance via elevated HBP flux.

**Figure supplement 1—source data 1.** Raw data.

**Figure supplement 2.** Generation of different AN3-12 mESC lines stably overexpressing WT FLAG-HA-hAMDHD2 and mutant FLAG-HA-hAMDHD2 D294A protein.

**Figure supplement 2—source data 1.** Raw data.

**Figure supplement 3.** Generation of AMDHD2 KO founder mice.

A comparison between both monomers from the dimer in the crystal revealed no major structural differences between monomer A and monomer B (*Figure 3—figure supplement 4*). The structure of the deacetylase domain showed a TIM (triosephosphate isomerase) barrel-like fold (*Figure 3D*). A typical TIM-barrel has eight alternating β-strands and α-helices forming a barrel shape where the parallel β-sheet builds the core that is surrounded by the α-helices. In AMDHD2, the eight alternating β-strands/α-helices are interrupted after eight β-strands and seven α-helices by an insertion of three antiparallel β-strands (β15–β17), which form an additional β-sheet close to the active site (*Figure 3C*, *Figure 3—figure supplement 5*). In monomer B, this β-sheet shows the highest crystallographic B-factors within the structure (*Figure 3—figure supplement 2*), indicating high flexibility and suggesting a functional role as a lid to the active site. The DUF-domain consists of two β-sheets, which are composed of three or six antiparallel β-strands each, and two small α-helices (*Figure 3D*). Taken together, these β-sheets form a β-sandwich. A superposition of the Zn-bound and the GlcN6P- and Zn-bound structures of AMDHD2 indicated no structural changes by the binding of the product (*Figure 3—figure supplement 6*). Residues from both monomers contribute to GlcN6P-binding (*Figure 3E*, *Figure 3—figure supplement 7A*). The phosphate group of the sugar is interacting via hydrogen bonds with Asn235 and Ala236, as well as ionic interactions to His242* and Arg243* of the other monomer (*Figure 3E*, *Figure 3—figure supplement 7A, B*). To assess a functional role of the residues His242* and Arg243*, and especially of the dimeric state on catalytic activity of AMDHD2, we generated the double mutant H242A/R243A and the mutants I280E and I280R, whose side chains might disrupt dimerization (*Figure 3F*). Analytical SEC measurements confirmed the presence of monomeric I280E (45.7±0.1 kDa) and monomeric I280R (44.4±0.5 kDa) compared to dimeric WT AMDHD2 (89.2±0.9 kDa) (*Figure 3G and H*). In contrast, the H242A/R243A substitution did not clearly disrupt dimerization (79.4±0.8 kDa) (*Figure 3G and H*). Strikingly, I280E, I280R, and H242A/R243A showed no catalytic activity, supporting that AMDHD2 must form a dimer to be active and that the residues His242* and Arg243* are indispensable for catalytic activity. GlcN6P binding to the active site is further mediated by hydrogen bonds between the hydroxyl groups of GlcN6P with Ala154 and His272. The catalytic Zn ion is coordinated via electrostatic interactions with Glu143, His211, His232, and two water molecules, which in turn are stabilized by interactions with GlcN6P and several amino acid side chains including Asp294 that might, based on the homology to bacterial NagA, act as the catalytic base (*Hall et al., 2007*; *Figure 3E*, *Figure 3—figure supplement 7A, B*). We confirmed the presence of a single Zn ion in the human AMDHD2 active site by measuring an anomalous signal at the Zn-K edge (*Figure 3J*). Given the conservation of all functional residues (*Figure 3—figure supplement 8*), the human AMDHD2 reaction mechanism is likely to be very similar to the proposed mechanism for the enzyme from *Escherichia coli* (*Hall et al., 2007*). In addition to GlcNAc-6P, bacterial NagAs are reported to use N-acetylgalactosamine-6-phosphate (GalNAc-6P) and N-acetylglucosamine-6-sulphate (GlcNAc-6S) as substrates, albeit with increased $K_m$ values (*Hall et al., 2007*; *Ahangar et al., 2018*). The high structural conservation of the side chains interacting with the sugar's C4 for GalNAc-6P or the phosphate group prompted us to test whether human AMDHD2 can catalyze the deacetylation of GalNAc-6P and GlcNAc-6S as well. Of note, we did not observe activity toward these

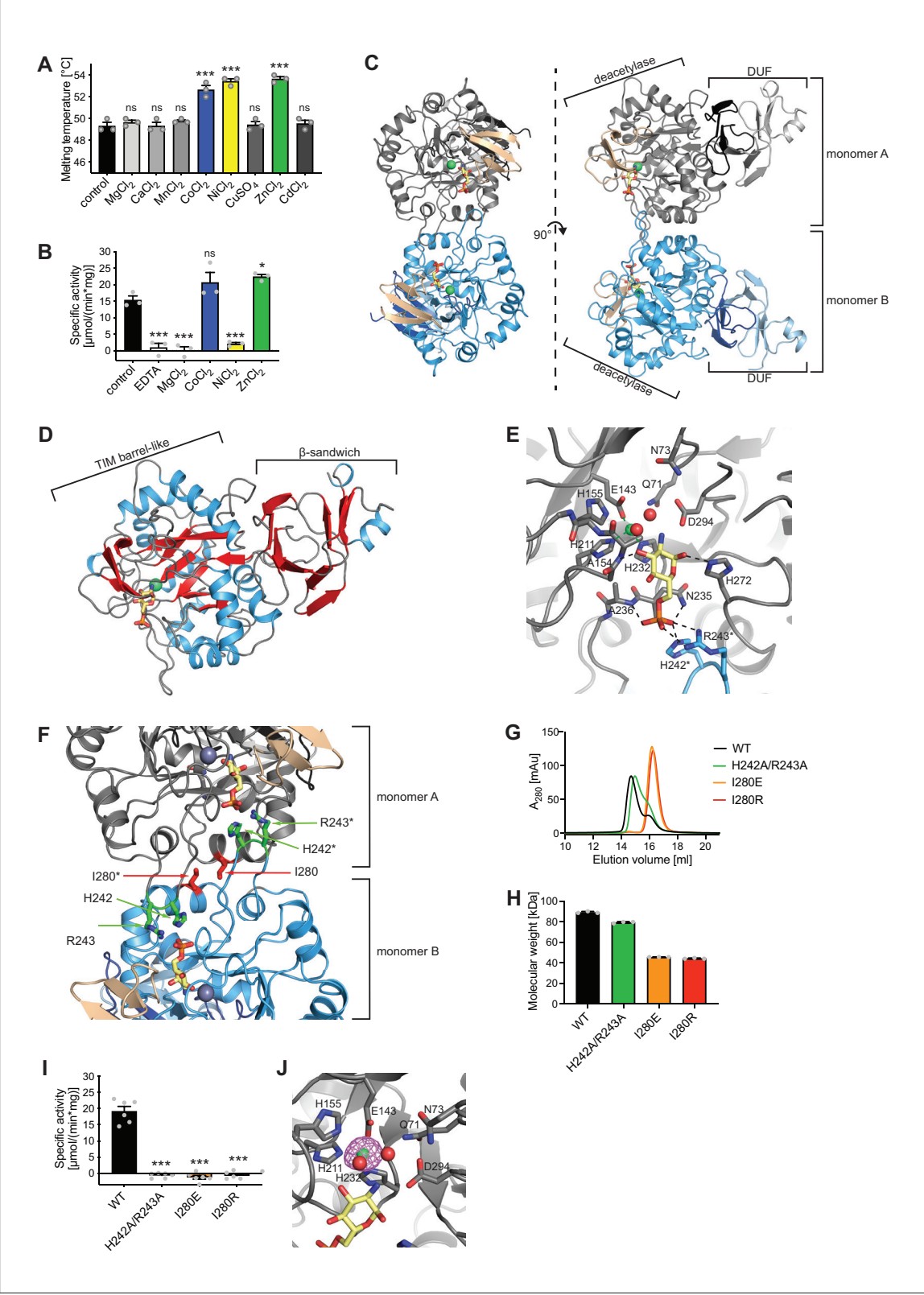

**Figure 3.** Structural and biochemical characterization of human AMDHD2. (**A**) Influence of divalent addition (10 µM) on the stability of AMDHD2 in size-exclusion chromatography (SEC) buffer in thermal shift assays (mean + SEM, n=3, ns=not significant, ***p<0.0001 versus wild-type (WT), one-way ANOVA). (**B**) Deacetylase activity of AMDHD2 in the presence of EDTA and several indicated divalents (mean + SEM, n=3, ns=not significant, *p<0.05, ***p<0.0001 versus WT, one-way ANOVA). (**C**) Overview of the human AMDHD2 dimer in cartoon representation. Monomer A is colored in gray and

*Figure 3 continued on next page*

*Figure 3 continued*

monomer B in blue. The two deacetylase domains are interacting with each other. The DUF domain is formed by residues of the N-terminus (light gray and light blue) and residues of the C-terminus (black and dark blue). GlcN6P (yellow sticks), $Zn^{2+}$ (green sphere), and the putative active site lid (wheat) are highlighted. (**D**) Domains and secondary structure elements within one AMDHD2 monomer. The deacetylase domain (left) shows a TIM barrel-like fold, while the small DUF domain (right) is composed of a β-sandwich fold. α-helices are colored in blue, β-strands in red, and loops in gray. GlcN6P (yellow sticks) and $Zn^{2+}$ (green sphere) are highlighted. (**E**) Close-up view of the active site in cartoon representation. Residues involved in ligand binding or catalysis are highlighted as sticks, as well as GlcN6P (yellow sticks), $Zn^{2+}$ (green sphere), and two water molecules (red spheres). The GlcN6P binding site is formed by two deacetylase domains. Black dashed lines indicate key interactions to GlcN6P and green dashed lines the coordination of $Zn^{2+}$. (**F**) Close-up view of the dimer interface in cartoon representation. His242, Arg243, and Ile280, which were mutated for further characterization of the dimer, are highlighted as sticks. (**G**) Representative chromatogram of an analytical SEC of human AMDHD2 variants using a Superdex 200 Increase 10/300 GL column. Absorption at 280 nm (mAU: milli absorbance units) was plotted against the elution volume. (**H**) Molecular weight of human AMDHD2 based on analytical SEC measurements (mean + SD, n=3.) (**I**) Deacetylase activity of wild-type (WT) and mutant human AMDHD2 (mean + SEM, n=6, ***p<0.0001 versus WT, one-way ANOVA). (**J**) Anomalous map of $Zn^{2+}$ with a contour level of 5.0 RMSD (violet).*Figure 3—source data 1*

The online version of this article includes the following source data and figure supplement(s) for figure 3:

**Source data 1.** Raw data.

**Figure supplement 1.** Structural and biochemical characterization of human AMDHD2.

**Figure supplement 2.** B-factor representation of WT human AMDHD2.

**Figure supplement 3.** Oligomeric state of human AMDHD2.

**Figure supplement 3—source data 1.** Raw data.

**Figure supplement 4.** Superposition of GlcN6P-bound AMDHD2 monomer A (gray) and monomer B (blue) in cartoon representation.

**Figure supplement 5.** Close-up view of the active site in cartoon representation.

**Figure supplement 6.** Superposition of the structures of GlcN6P-bound (gray and blue) and GlcN6P-free (green and red) human AMDHD2 with RMSD of 0.67 Å over 792 main chain residues in cartoon representation.

**Figure supplement 7.** Active site of human AMDHD2.

**Figure supplement 8.** Protein sequence alignment of AMDHD2.

**Figure supplement 9.** Deacetylase activity of human AMDHD2 toward GlcNAc6P, GalNAc6P, and GlcNAc6S (mean + SEM, n=6, ***p<0.0001 versus GlcNAc6P, one-way ANOVA).

**Figure supplement 9—source data 1.** Raw data.

N-acetyl amino sugars that might be of physiological relevance (*Figure 3—figure supplement 9*). In summary, our data show that human AMDHD2 is an obligate dimeric protein with high specificity for GlcNAc-6P that carries a single catalytic Zn ion in the active center.

## Characterization of AMDHD2 LOF mutants

We next characterized the 11 AMDHD2 substitutions from our screen and the putative active site mutant D294A to understand how they might affect the function of AMDHD2. Many AMDHD2 variants were soluble upon bacterial expression, including F146L, A154P, T185A, S208T, and D294A (*Figure 4A*). These substitutions are located close to the active site of AMDHD2 (*Figure 4B*) and Ala154 is involved in ligand binding by donating an H-bond via its main chain NH group to the 3-OH group of the sugar (*Figure 3E*, *Figure 3—figure supplement 7A, B*). In contrast, no soluble expression could be achieved for AMDHD2 G102D, G130R, G226E, and G265V (*Figure 4A*, *Figure 4—figure supplement 1*). The substitution of the small, flexible glycine by charged and/or bigger residues are likely to be incompatible with the proper tertiary structure and/or the folding process, thus resulting in insoluble AMDHD2 protein variants that remain in the pellet fraction after sample lysis (*Figure 4—figure supplement 1*). The effect of the L142F mutation was even more severe as the substitution of Leu142 by the bigger phenylalanine resulted in AMDHD2 fragmentation (*Figure 4A*). Also, the I38T and G265R substitutions reduced soluble expression, indicating disturbed protein folding. We next tested the consequences of the I38T, T185A, G265R, and D294A substitutions on AMDHD2 activity. AMDHD2 T185A showed reduced activity and no activity was detected for G265R and D294A, while the third substitution, I38T, remained active (*Figure 4C*). This result indicates a functional role of Asp294 in the catalytic mechanism, confirming our idea that this substitution inactivates AMDHD2 and justifying its use in the rescue experiments (*Figure 2C and D*). Asp294 is likely to act as catalytic base that activates the nucleophilic water molecule together with the metal ion, and later protonating the leaving group (*Hall et al., 2007*). Moreover, the I38T substitution is the only identified mutation

**Table 1.** Data collection and refinement statistics of human AMDHD2.

| | AMDHD2+Zn+GlcN6P | AMDHD2+Zn |
|---|---|---|
| Wavelength (Å) | 1.00 | 1.00 |
| Resolution range (Å) | 45.71–1.90 (1.97–1.90) | 48.21–1.84 (1.90–1.84) |
| Space group | P $2_1 2_1 2_1$ | P $2_1 2_1 2_1$ |
| a, b, c (Å) | 63.3, 161.4, 86.6 | 61.8, 84.3, 154.2 |
| α, β, γ (°) | 90, 90, 90 | 90, 90, 90 |
| Total reflections | 428,727 (42,693) | 468,961 (46,539) |
| Unique reflections | 70,760 (6907) | 71,036 (6953) |
| Multiplicity | 6.1 (6.2) | 6.6 (6.7) |
| Completeness (%) | 99.7 (98.3) | 99.9 (99.2) |
| Mean I/sigma(I) | 11.46 (1.16) | 12.53 (1.06) |
| Wilson B-factor | 34.7 | 29.7 |
| $R_{merge}$ (%) | 9.5 (140.4) | 10.2 (150.6) |
| $R_{meas}$ (%) | 10.4 (153.3) | 11.0 (163.3) |
| $R_{pim}$ (%) | 4.2 (60.9) | 4.3 (62.5) |
| $CC_{1/2}$ (%) | 99.9 (49.4) | 99.9 (49.9) |
| CC* (%) | 100 (81.3) | 100 (81.6) |
| Reflections used in refinement | 70,751 (6906) | 71,024 (6952) |
| Reflections used for R-free | 1980 (194) | 1992 (195) |
| $R_{work}$ (%) | 18.5 (31.7) | 18.2 (31.2) |
| $R_{free}$ (%) | 21.3 (29.6) | 20.6 (33.6) |
| $CC_{work}$ (%) | 96.6 (73.4) | 96.6 (75.7) |
| $CC_{free}$ (%) | 94.4 (73.1) | 95.4 (72.8) |
| Number of non-hydrogen atoms | 6361 | 6331 |
| Macromolecules | 5997 | 5999 |
| Ligands | 34 | 14 |
| Solvent | 330 | 318 |
| Protein residues | 801 | 798 |
| RMS (bonds) (Å) | 0.005 | 0.004 |
| RMS (angles) (°) | 0.69 | 0.70 |
| Ramachandran favored (%) | 96.9 | 97.1 |
| Ramachandran allowed (%) | 2.9 | 2.7 |
| Ramachandran outliers (%) | 0.25 | 0.25 |
| Rotamer outliers (%) | 0.32 | 0.32 |
| Clashscore | 0.67 | 0.92 |
| Average B-factor | 43.59 | 37.67 |
| Macromolecules | 43.81 | 37.76 |
| Ligands | 40.19 | 41.98 |
| Solvent | 40.10 | 35.72 |
| Number of TLS groups | 4 | 4 |
| PDB ID | 7NUT | 7NUU |

Statistics for the highest-resolution shell are shown in parentheses.

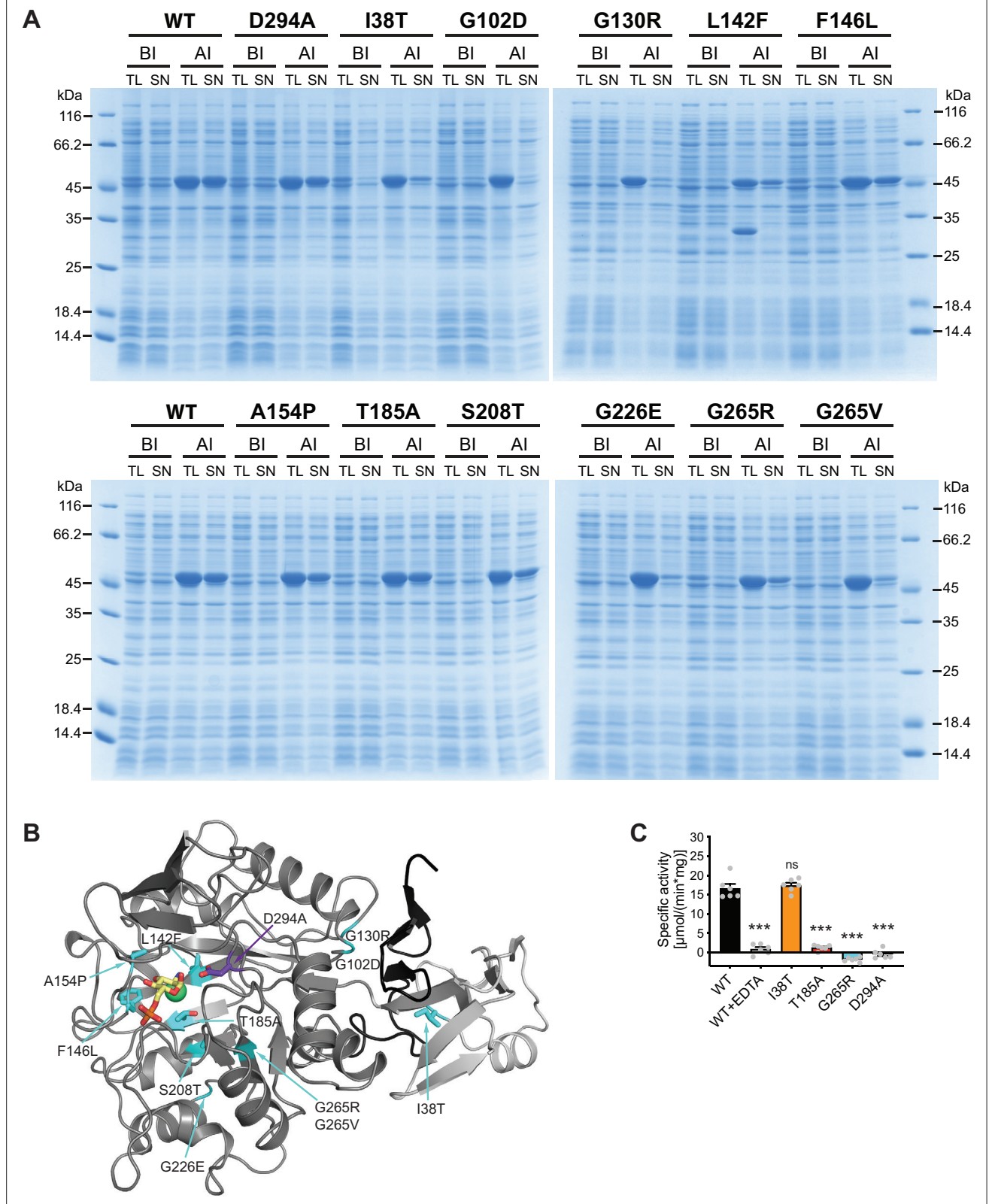

**Figure 4.** Characterization of AMDHD2 loss-of-function (LOF) mutants. (**A**) SDS-gels stained with Coomassie brilliant blue of a representative bacterial test expression of the human AMDHD2 variants. The experiment was repeated three times with similar results. BI: before induction, AI: after induction, TL: total lysate, SN: soluble fraction/supernatant. A band corresponding to the molecular weight of human AMDHD2 with His$_6$-tag (46 kDa) was present in all total lysates after induction. (**B**) Overview of the position of the potential LOF mutations in human AMDHD2 in cartoon representation. GlcN6P

*Figure 4 continued on next page*

Figure 4 continued

(yellow sticks), the metal co-factor (green spheres), the active site Asp294 (violet sticks), and the 11 putative LOF mutations (cyan sticks) are highlighted. (C) Deacetylase activity of wild-type (WT) and mutant human AMDHD2 (mean + SEM, n=6, ***p<0.0001 versus WT, one-way ANOVA). **Figure 4—source data 1**

The online version of this article includes the following source data and figure supplement(s) for figure 4:

**Source data 1.** Raw data.

**Figure supplement 1.** Characterization of AMDHD2 loss-of-function mutants.

**Figure supplement 1—source data 1.** Raw data.

from the screen that is located in the DUF domain of AMDHD2. It reduced bacterial AMDHD2 expression yields, suggesting impaired protein folding. This is likely to result in a LOF in vivo, while the purified and soluble protein is active. Taken together, the structural and biochemical characterization of AMDHD2 revealed that LOF and subsequent HBP activation resulted from folding defects in AMDHD2 or it was caused by a loss of catalytic activity.

## AMDHD2 limits HBP activity when GFPT2 replaces GFPT1 as the first enzyme

Having established that a loss of AMDHD2 function results in HBP activation, we wondered about the role of the HBP's rate-limiting enzyme GFPT1. Under normal conditions, GFPT1 is constantly feedback inhibited by UDP-GlcNAc, crucially limiting HBP activity (**Ruegenberg et al., 2020**). A GOF substitution in GFPT1 (G451E), however, increased HBP flux in nematodes and in murine cells, demonstrating a high degree of conservation (**Horn et al., 2020**). In AN3-12 cells, the G451E GOF substitution, introduced into the genomic locus by CRISPR/Cas9, as well as a *Gfpt1* KO did not affect UDP-GlcNAc levels (**Figure 5—figure supplement 1**). While *Gfpt1* is widely expressed across cell types, it is known that in some tissues *Gfpt2* is the predominantly expressed paralog (**Oki et al., 1999**). Since loss of GFPT1 did not affect HBP activity, we hypothesized that GFPT2 instead of GFPT1 might control metabolite entry into the HBP in AN3-12 mESCs. Indeed, *Gfpt2*, mRNA was abundantly expressed in AN3-12 cells, while expression levels of *Gfpt1* were comparatively low (**Figure 5A**). Next, we performed Western blot analysis using pure purified human GFPT and compared those to the GFPT abundance in various cell lines. GFPT2 was found abundantly expressed, while GFPT1 was difficult to detect in AN3-12 mESCs (**Figure 5B**). E14 mESCs likewise showed predominant GFPT2 expression and low GFPT1 abundance. In contrast, mouse neuronal N2a cells as well as muscle precursor C2C12 myoblasts showed predominant GFPT1 expression and GFPT2 was virtually undetectable. These data suggest an HBP configuration characterized by a high GFPT2:GFPT1 ratio in mESCs.

To further investigate the possibility of ESC-specific HBP regulation, we next checked AMDHD2 levels in mESCs and compared them to cells using GFPT1 as the predominant first HBP enzyme. Mirroring GFPT2 levels, AMDHD2 protein abundance was higher in mESCs compared to N2a and C2C12 cells (**Figure 5C**). Moreover, the KO of AMDHD2 in AN3-12 mESCs resulted in a drastic elevation of UDP-GlcNAc levels, while the loss of AMDHD2 in N2a cells had no significant effect (**Figure 5D**). In accordance, the loss of AMDHD2 in C2C12 myoblasts was not sufficient to increase UDP-GlcNAc levels compared to control cells (**Figure 5—figure supplement 2A-B**). This indicates that AMDHD2 was constitutively active in AN3-12 cells, while the catalysis of the reverse flux of the HBP by AMDHD2 seemed to be negligible in N2a and C2C12 cells. We therefore hypothesized that AMDHD2 plays a key role in the HBP when GFPT2 is its first enzyme instead of the more common GFPT1. Our previous data indicate that GFPT1 is under constant UDP-GlcNAc inhibition, sufficient for full suppression of GFPT1 activity (**Ruegenberg et al., 2020**). We reasoned that higher UDP-GlcNAc levels in mESCs can only be achieved by differences in UDP-GlcNAc feedback inhibition between GFPT1 and GFPT2. To address this point, we generated recombinant human GFPT1 and GFPT2 with an internal His$_6$tag and characterized the proteins in activity assays (**Figure 5E**, **Table 2**, **Figure 5—figure supplement 3A-B**). Kinetic measurements confirmed that both proteins were fully functional and revealed different substrate affinities of GFPT2 compared to GFPT1 (**Table 2**, **Figure 5—figure supplement 3A-B**). In a UDP-GlcNAc dose-response assay, we found a significantly higher IC$_{50}$ value for GFPT2 (367.3–43.6/+49.5 μM) compared to GFPT1 (57.0–8.3/+9.7 μM) (**Figure 5E**, **Table 2**). We conclude, first, that UDP-GlcNAc inhibition is weaker in GFPT2 compared to GFPT1 and, second, that AMDHD2

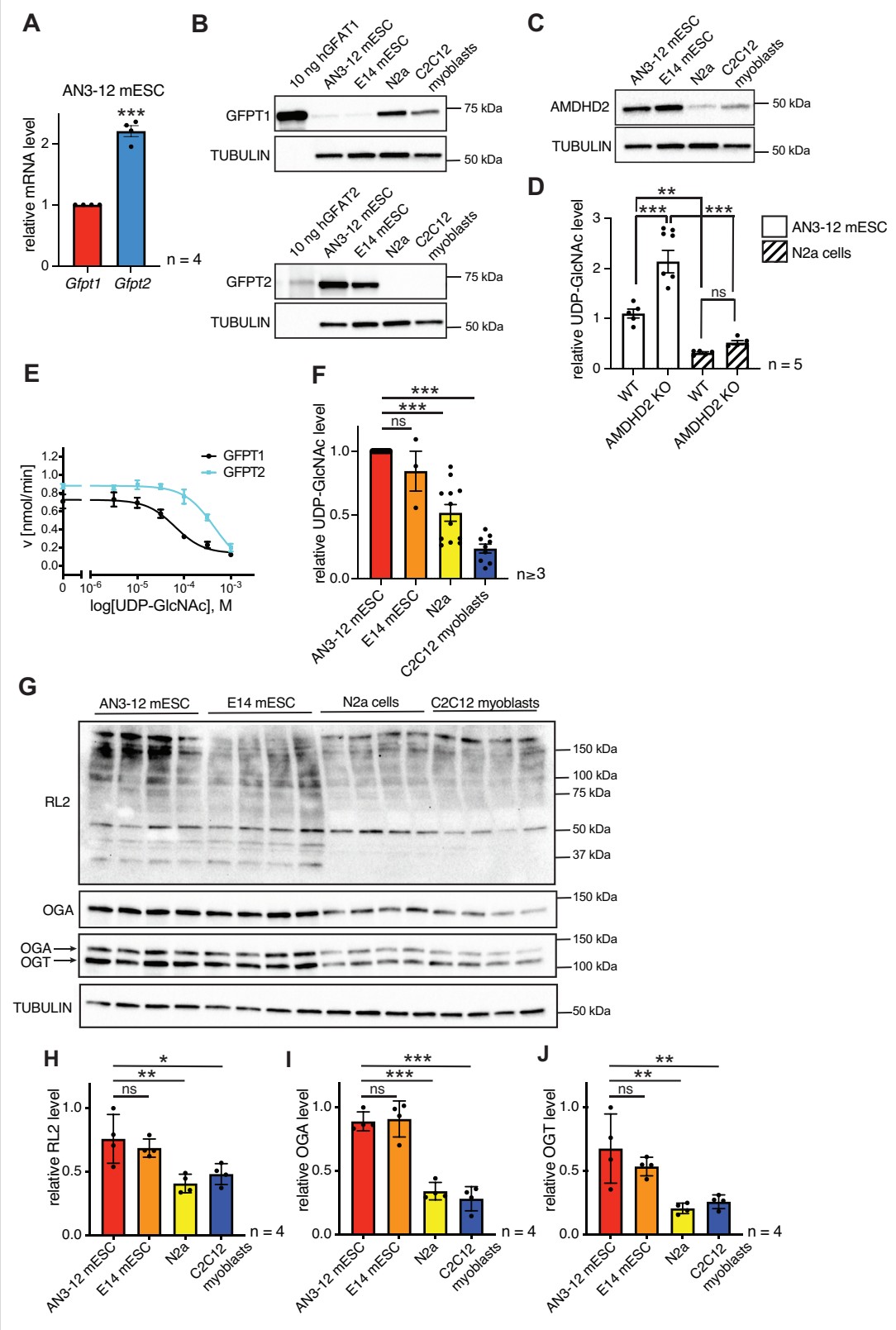

**Figure 5.** AMDHD2 limits HBP activity when GFPT2 replaces GFPT1 as the first enzyme. (**A**) Relative *GFPT1* and *GFPT2* mRNA levels (qPCR) in WT AN3-12 cells (mean ± SEM, n=4, ***p<0.001, unpaired t-test). (**B**) Western blot analysis of purified human GFPT1 and GFPT2 protein lysates of indicated cell lines. (**C**) Western blot analysis of AMDHD2 in indicated cell lines. (**D**) IC-MS analysis of UDP-GlcNAc levels in WT and AMDHD2 KO AN3-12 mESCs and N2a cells (mean ± SEM, n=5, ***p<0.001, one-way ANOVA). (**E**) Representative UDP-GlcNAc dose-response assay with hGFPT1 (black circle) and

*Figure 5 continued on next page*

*Figure 5 continued*

hGFPT2 (teal square) (mean ± SD, n=3). (**F**) Relative UDP-GlcNAc levels in indicated cell lines measured by IC-MS. Levels are normalized to those in AN3-12 mESCs (mean ± SEM, n≥3, ***p< 0.001, one-way ANOVA). (**G**) Western blot analysis of O-GlcNAc-modified proteins (RL2), OGA, OGT, and tubulin in the indicated cell lines. (**H–J**) Quantification of Western blot in (**G**) (mean ± SD, n=4, *p<0.05, **p<0.01, ***p<0.001, ns=not significant, one-way ANOVA Dunnett's posttest).***Figure 5—source data 1*** HBP, hexosamine biosynthetic pathway; WT, wild-type.

The online version of this article includes the following source data and figure supplement(s) for figure 5:

**Source data 1.** Raw data.

**Figure supplement 1.** AMDHD2 limits HBP activity when GFPT2 replaces GFPT1 as the first enzyme.

**Figure supplement 1—source data 1.** Raw data.

**Figure supplement 2.** Deletion of AMDHD2 has no effect on HBP flux in C2C12 myoblasts.

**Figure supplement 2—source data 1.** Raw data.

**Figure supplement 3.** Biochemical characterization of human GFPT2 compared to human GFPT1.

**Figure supplement 3—source data 1.** Raw data.

plays a crucial role in balancing GFPT2-mediated HBP flux. Consistent with lower feedback inhibition of GFPT2, UDP-GlcNAc levels in AN3-12 and E14 mESCs were significantly higher than in N2a and C2C12 cells with a GFPT1-regulated HBP (***Figure 5F***). We also tested protein O-GlcNAc modification, which relies on UDP-GlcNAc as a precursor molecule, in the different cell lines via Western blot analysis with an O-GlcNAc specific antibody (RL2). Consistent with the elevated UDP-GlcNAc levels, we observed significantly higher O-GlcNAc modification in mESCs. Of note, both OGA and OGT were more abundant in the mESCs compared to N2a and C2C12 cells (***Figure 5G–J***). Overall, these data demonstrate an mESC-specific configuration of the HBP, relying on the coexpression of GFPT2 and AMDHD2. This balance appears to be tuned to elevate UDP-GlcNAc levels and O-GlcNAc modification in ESCs.

## Differentiation of ESCs induces an enzymatic reconfiguration of the HBP by reducing the GFPT2:GFPT1 ratio

In the next step, we asked if differentiation of mESC might affect the HBP's enzymatic configuration. For this, we removed leukemia inhibitory factor (LIF) from the medium, initiating differentiation (***Hocke et al., 1995***). LIF removal for 5 days resulted in partial differentiation of AN3-12 cells as indicated by a decrease of stem cell markers (***Figure 6—figure supplement 1A***). Of note, GFPT2 protein as well as *Gfpt2* mRNA levels decreased significantly with LIF removal (***Figure 6A and B***). GFPT1 and AMDHD2 mRNA and protein levels did not change in this partial differentiation paradigm (***Figure 6—figure supplement 1B-D***). A decrease in the GFPT2:GFPT1 ratio upon differentiation was also observed in published data sets: relative GFPT2 mRNA and protein levels decrease during neuronal differentiation (***Saez et al., 2018***) and during the differentiation in the cardiac lineage (***Frank et al., 2019***; ***Bartsch et al., 2021***) in human ESCs (***Figure 6C and D***).

## Discussion

HBP activation increases cellular UDP-GlcNAc levels that protect from TM toxicity (***Denzel et al., 2014***). We used this knowledge to interrogate the HBP for additional regulators in a forward genetic TM resistance screen using haploid mammalian cells. Random chemical DNA mutagenesis at high

**Table 2.** Kinetic parameters of human GFPT1 and GFPT2.

| | L-Glu production | | | D-GlcN6P production | | | UDP-GlcNAc inhibition |
|---|---|---|---|---|---|---|---|
| | $K_m$ L-Gln [mM] | $k_{cat}$ [s$^{-1}$] | $k_{cat}/K_m$ [mM$^{-1}$ s$^{-1}$] | $K_m$ Frc6P [mM] | $k_{cat}$ [s$^{-1}$] | $k_{cat}/K_m$ [mM$^{-1}$ s$^{-1}$] | IC$_{50}$ [μM] |
| GFPT1 | 1.1±0.19 | 3.6±0.18 | 3.3 | 0.08±0.01 | 1.7±0.09 | 21.3 | 57.0–8.3/+9.7 |
| GFPT2 | 0.5±0.06 | 3.7±0.10 | 7.4 | 0.29±0.05 | 1.8±0.09 | 6.2 | 367.3–43.6/+49.5 |
| Unpaired t-test (two-sided) | **p=0.005 | | | **p=0.0027 | | | ***p=0.0002 |

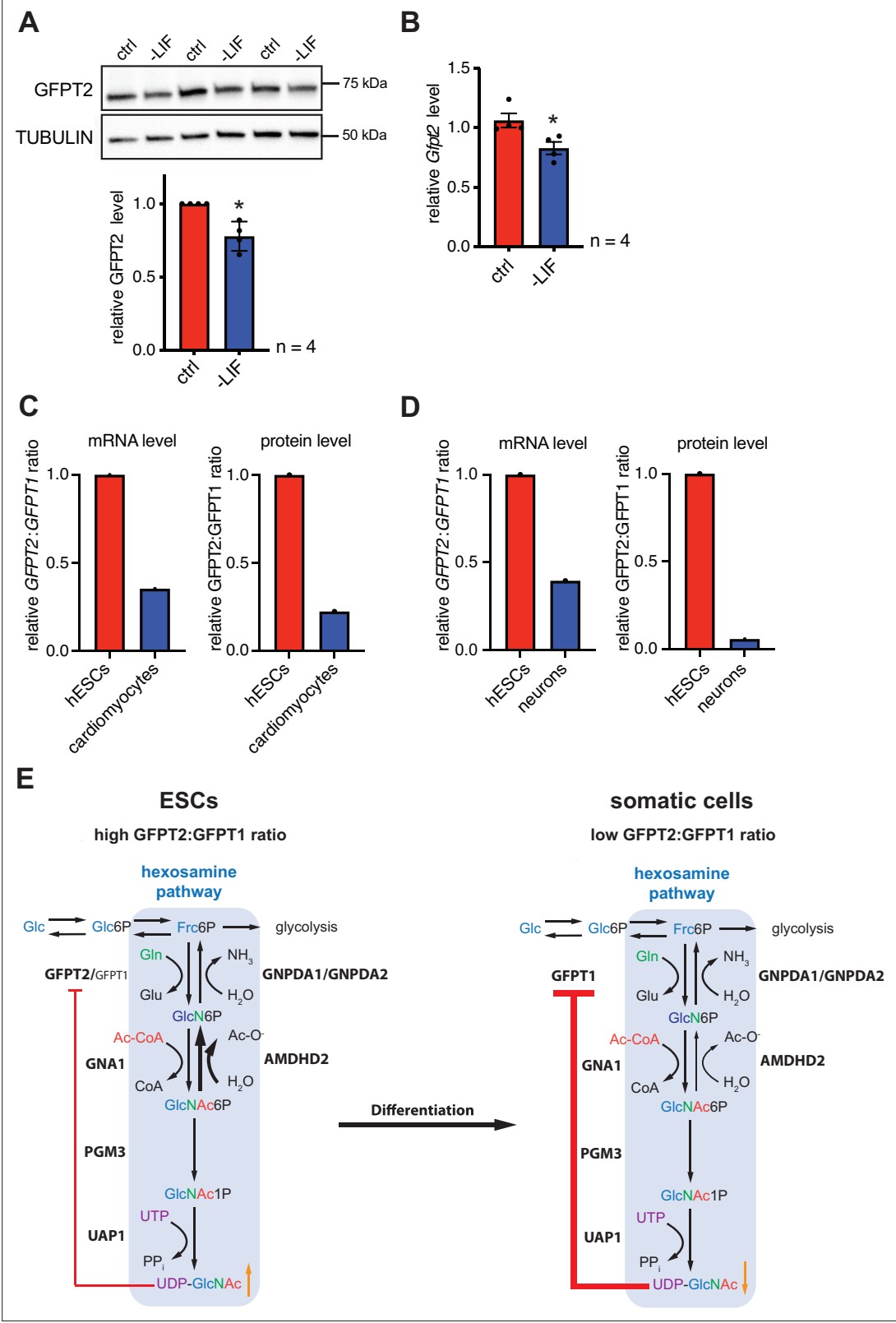

**Figure 6.** Differentiation of ESCs induces an enzymatic reconfiguration of the HBP by reducing the GFPT2:GFPT1 ratio. (**A**) Western blot analysis and quantification (mean ± SD, n=4, *p<0.05, unpaired t-test) of GFPT2 in WT AN3-12 control cells and upon partial differentiation by a 5-day LIF removal. (**B**) Relative *Gfpt2* mRNA level (qPCR) in WT AN3-12 cells and upon partial differentiation by a 5-day LIF removal (mean ± SEM, n=4, *p<0.05, unpaired t-test). (**C**) Relative *GFPT2/GFPT1* mRNA and GFPT2/GFPT1 protein ratios in human ESCs and upon differentiation into cardiomyocytes

*Figure 6 continued on next page*

*Figure 6 continued*

(data obtained from: *Frank et al., 2019*; *Bartsch et al., 2021*). (**D**) Relative *GFPT2/GFPT1* mRNA and GFPT2/GFPT1 protein ratios in human ESCs and upon differentiation into neurons (data obtained from: *Saez et al., 2018*). (**E**) Model: enzymatic configuration of the HBP in ESCs and somatic cells. The HBP (blue box) generates UDP-GlcNAc in multiple enzymatic steps. While ESCs mainly rely on GFPT2, more differentiated cells use GFPT1 for HBP entry. GFPT2 is less susceptible to UDP-GlcNAc inhibition than GFPT1 (indicated by red arrow). As an alternative regulatory mechanism ESCs require AMDHD2. Differentiation of ESCs induces an HBP reconfiguration, resulting in a decreased GFPT2:GFPT1 ratio. GFPT: glutamine fructose-6-phosphate amidotransferase, GNPDA: D-glucosamine-6-phosphate deaminase, GNA1: D-glucosamine-6-phosphate-N-acetyltransferase, AMDHD2: N-acetylglucosamine-6-phosphatedeacetylase, PGM3: phosphoglucomutase, UAP1: UDP-N-acetylglucosamine pyrophosphorylase.*Figure 6—source data 1* ESC, embryonic stem cell; HBP, hexosamine biosynthetic pathway; LIF, leukemia inhibitory factor; WT, wild-type.

The online version of this article includes the following source data and figure supplement(s) for figure 6:

**Source data 1.** Raw data.

**Figure supplement 1.** Differentiation of ESCs induces an enzymatic reconfiguration of the HBP by reducing the GFPT2:GFPT1 ratio.

**Figure supplement 1—source data 1.** Raw data.

saturation in haploid cells is a unique strategy to identify recessive mutations including those leading to single amino acid substitutions. Using this approach, we identified the N-acetylglucosamine-6-phosphate deacetylase AMDHD2 as a novel regulator of the mammalian HBP. With an independent random insertional mutagenesis screen, we confirmed the importance of AMDHD2 for regulating HBP activity and confirmed the role of AMDHD2 through rescue experiments. We next solved the first crystal structure of human AMDHD2 and noted that resistance-associated substitutions disturb protein folding or cluster in the catalytic pocket, likely interfering with substrate binding or catalysis. Finally, we found that mESCs utilize GFPT2 for metabolite entry into the HBP instead of the more widely expressed GFPT1. GFPT2 is under considerably reduced UDP-GlcNAc feedback inhibition explaining why loss of AMDHD2 activity was sufficient for HBP activation without GFPT mutations (*Figure 6E*).

Chemical mutagenesis-based screening in haploid cells represents a powerful and unique technique. This approach allows to dissect the entire spectrum of mutations including LOF, GOF, and neomorph alleles, and at the same time allows structure-function analyses due to its amino acid resolution (*Horn et al., 2018*). The additional use of haploid cells not only enables detection of dominant but also recessive mutations due to the lack of a remaining and interfering WT allele. Of note, identification of AMDHD2 as a novel regulator of the HBP was only possible in this specific setup since *Amdhd2* mutations are recessive as shown in the AMDHD2 KO mouse.

Besides the function as GlcNAc-6P deacetylase, AMDHD2 was shown to be involved in the degradation of N-glycolylneuraminic acid (Neu5Gc) in mice and in human cell culture (*Bergfeld et al., 2012*; *Campbell et al., 1987*). Nevertheless, mammalian AMDHD2 is rather unstudied and most knowledge is based on the bacterial homolog NagA. NagA catalyzes the deacetylation reaction in the HBP, contributing to recycling of cell wall components such as GlcNAc. Since breakdown of GlcNAc can be used as an energy source by bacteria and fungi, NagA plays a crucial role in their energy metabolism (*Liu et al., 2013*; *Plumbridge, 2009*; *Popowska et al., 2012*; *Yadav et al., 2011*). For this reason, HBP enzymes are attractive selective targets for antifungal and antibiotic drugs (*Zhang et al., 2021*; *Świątek et al., 2012a*; *Świątek et al., 2012b*; *Lockhart et al., 2020*). While catabolism of amino sugars connects GlcNAc with other important metabolic pathways, AMDHD2 had not been implicated in a regulatory role of the HBP and cellular UDP-GlcNAc homeostasis.

After identification of AMDHD2 as a key modulator of the mammalian HBP, we structurally and biochemically characterized human AMDHD2. We solved the structure of human AMDHD2, the first reported eukaryotic structure of an AMDHD2 homolog and confirmed that human AMDHD2 is an obligate dimeric enzyme. Residues from both monomers contribute to ligand binding in the active site, while the residues important for catalysis originate from one monomer. The oligomeric state of AMDHD2 is therefore a plausible target to modulate its catalytic properties.

We showed that the mutations identified in the screen cause a LOF in human AMDHD2 by disrupting its folding or activity (*Figure 4*). AMDHD2 is composed of a deacetylase domain and a small DUF. We identified only one mutation, I38T, within the DUF domain and this mutant showed diminished expression yields and low solubility, potentially explaining the LOF. Nonetheless, the soluble fraction of AMDHD2 I38T was as active as WT AMDHD2 in activity assays, indicating that the DUF domain might be dispensable for catalysis.

Further characterizing the HBP, we noticed a surprising configuration of HBP enzymes in AN3-12 and E14 mESCs. While N2a cells and C2C12 myoblasts rely on GFPT1 as the key HBP enzyme, the mESCs use GFPT2 that is abundantly expressed (*Figure 6E*). Consistently, genetic manipulation of GFPT1 did not show any effect on UDP-GlcNAc levels in AN3-12 mESCs, while introducing the G451E GOF mutation in GFPT1 of N2a cells leads to the previously reported boost of HBP activity (*Horn et al., 2020*). Additionally, AMDHD2 abundance was higher in mESCs (*Figure 5C*). In accordance, the AMDHD2 KO in AN3-12 mESCs massively elevated UDP-GlcNAc levels, while the loss of AMDHD2 in N2a cells and C2C12 myoblasts had no significant impact. Under physiological conditions, GFPT1 is strongly inhibited by UDP-GlcNAc (*Ruegenberg et al., 2020*). In this scenario, as is the case in N2a and C2C12 cells, loss of the reverse flux by AMDHD2 KO showed no drastic effect on UDP-GlcNAc levels (*Figure 6E*). Moreover, we showed that GFPT2 has altered substrate affinities and is less susceptible to UDP-GlcNAc feedback inhibition. N- or C-terminal tags in GFPT disturb the catalytic function, therefore the GFPT preparations used here carry an internal tag for purification at a position that is reported not to interfere with the kinetic properties of GFPT1 (*Richez et al., 2007*). Studies with other tagging strategies reported only a weak inhibition of GFPT2 by UDP-GlcNAc (*Hu et al., 2004*; *Oliveira et al., 2021*). In contrast, we demonstrate that GFPT2 can be fully inhibited by UDP-GlcNAc with an approximately six-fold higher $IC_{50}$ value compared to GFPT1. Overall, our data suggest that GFPT1 is sufficiently regulated by feedback inhibition to determine HBP flux under physiological conditions. Cells using GFPT2 in the HBP, in contrast, rely on AMDHD2 to balance forward and reverse flux in the HBP.

This HBP configuration might be a general adaptation of mESCs as we could show similar results for AN3-12 and E14 mESCs. Differentiation might result in a switch of GFPT expression and indeed partial differentiation of AN3-12 cells by LIF removal induced a significant decrease in GFPT2 levels. GFPT1 and AMDHD2 levels were not affected likely due to the early differentiation state. Analysis of published data confirmed that GFPT2 is highly expressed in human ESCs, and abundance decreased during neuronal or myocyte differentiation, indicating a conserved mechanism in human ESCs. Consistent with these findings, intestinal stem cells in *Drosophila melanogaster* likewise express GFPT2 (*Mattila et al., 2018*). One potential consequence of this metabolic adaptation in ESCs is a higher baseline UDP-GlcNAc concentration compared to cells that use GFPT1 to control the HBP. This increase in UDP-GlcNAc concentration might affect downstream PTMs, which in turn can influence cell signaling. In particular, O-GlcNAc modifications already have been linked to stemness and pluripotency (*Constable et al., 2017*; *Jang et al., 2012*). Indeed, we detected increased O-GlcNAc levels in mESCs compared to cells utilizing GFPT1 in the HBP. Of note, not only UDP-GlcNAc levels but also the two essential enzymes for O-GlcNAc cycling OGA/OGT where significantly increased in mESCs, indicating a multilayered mechanism of maintaining high O-GlcNAc levels in mESCs. Additional significance of an ESC-specific HBP configuration might come from an adaptation to their special nutrient and energy requirements. ESCs show a specialized metabolic profile that likely affect the concentrations of GFPT substrates (*Intlekofer and Finley, 2019*). The kinetic properties of GFPT2 might reflect an adaption to substrate availability in ESCs. Consistently, GFPT2 is also upregulated in other rapidly proliferating cells with similar metabolic profiles, like in various types of cancer cells (*Oikari et al., 2018*; *Shaul et al., 2014*; *Zhang et al., 2018*; *Szymura et al., 2019*).

Taken together, we identify AMDHD2 as a novel essential gene in embryonic development and describe a cell type-specific role of AMDHD2 acting in tandem with GFPT2 to regulate the HBP in ESCs. Tuning HBP metabolic activity is relevant in cellular stress resistance, oncogenic transformation, growth, and in age-related diseases as cancer, diabetes, cardiovascular diseases, or neurodegenerative diseases (*Marshall et al., 1991*; *Oikari et al., 2018*; *Arnold et al., 1996*; *Champattanachai et al., 2007*). Of note, eukaryotic AMDHD2 was barely characterized and the identification of its critical role in HBP regulation paves the way for novel approaches to tackle age-associated pathologies, among other potential interventions. Our work advances the understanding of HBP control and provides specific means to beneficially affect these processes in the future.

## Materials and methods

### Cell lines and culture conditions

AN3-12 mouse embryonic haploid stem cells were cultured as previously described (*Elling et al., 2011*). In brief, Dulbecco's modified Eagle's medium (DMEM) high glucose (Sigma-Aldrich) was supplemented with glutamine, fetal bovine serum (FBS; 15%), penicillin/streptomycin, nonessential amino acids, sodium pyruvate (all Thermo Fisher Scientific, Waltham, MA), β-mercaptoethanol, and LIF (both Merck Millipore) and used to culture cells at 37°C in 5% $CO_2$ on noncoated tissue culture plates. For partial differentiation of AN3-12 cells, cells were seeded at a density of 2000–3000 cells/6-well and incubated for 5 days in medium without LIF.

N2a mouse neuroblastoma cells (RRID:CVCL_0470) and C2C12 (RRID:CVCL_0188) cells were cultured in DMEM containing 4.5 g/L glucose (Gibco) supplemented with 10% FBS (Gibco) and penicillin/streptomycin at 37°C in 5% $CO_2$. No mycoplasma contamination was detected.

### Cell sorting

To maintain a haploid cell population, cells were stained with 10 µg/ml Hoechst 33342 (Thermo Fisher Scientific) for 30 min at 37°C. To exclude dead cells, propidium iodide (Sigma-Aldrich) staining was added. Cells were sorted for DNA content on a FACSAria Fusion sorter and flow profiles were recorded with the FACSDiva software (BD).

### Cell viability assay (XTT)

Relative cell viability was assessed using the XTT cell proliferation Kit II (Roche Diagnostics) according to the manufacturer's instructions. TM treatments were performed for 48 hr, starting 24 hr after cell seeding. XTT turnover was normalized to corresponding untreated control cells.

### ENU mutagenesis screen, exome sequencing, and analysis

The screening procedure and the data analysis were extensively described previously (*Horn et al., 2018*). In brief, AN3-12 mouse embryonic haploid stem cells were mutagenized with 0.01 mg/ml Ethylnitrosourea for 2 hr at room temperature prior to drug selection starting 24 hr post mutagenesis using 0.5 µg/ml TM (Merck Millipore). After 21 days of drug selection, resistant clones were isolated and subjected to TM cytotoxicity assays and gDNA extraction using the Gentra Puregene Tissue Kit (QIAGEN). Paired-end, 150-bp whole-exome sequencing was performed on an Illumina Novaseq 6000 instrument after precapture-barcoding and exome capture with the Agilent SureSelect Mouse All Exon Kit. For data analysis, raw reads were aligned to the reference genome mm9. Variants were identified and annotated using GATK (v.3.4.46) and snpEff (v.4.2). TM resistance causing alterations were identified by allelism only considering variants with moderate or high effect on protein and a read coverage >10.

### Retroviral-based insertional mutagenesis screen and integration site mapping

The generation of a comprehensive cell bank of haploid AN3-12 cells, containing insertions in 16,970 mouse genes, was already created and described elsewhere (*Elling et al., 2019*; https://www.haplobank.at/ecommerce/control/main). In short, for retroviral library generation, enhanced gene-trap (EGT) viruses carrying a neomycin-resistance cassette were packaged in PlatinumE (Cell Biolabs) cells. The virus was concentrated by centrifugation (25,000 rpm, 4°C, 4 hr) and haploid mESCs were infected for 8 hr in the presence of 2 µg/ml polybrene. Upon infection for 30 hr, cells were treated with 0.2 mg/ml G418 (Gibco) for selection of gene-trap insertions. To estimate numbers of integrations, 500,000 cells were plated on 15-cm dishes, selected for integrations using G418 selection and colonies counted after 10 days. For comparison, 5000 cells were plated without selection. From the barcoded AN3-12 Retro Library, 3 million cells were plated on 15-cm plates and drug selection was performed for 21 days starting 24-hr post mutagenesis using 0.5 µg/ml TM (Merck Millipore). Resistant clones were isolated and subjected to TM cytotoxicity assays. Mapping of the genomic integration site was performed by inverse PCR. The genomic region was amplified using the primers 'DS' and 'US' (primers are listed in *Supplementary file 1*). The PCR reaction was analyzed on an agarose gel,

purified, and used for Sanger Sequencing with primer 'DS.' Sequences were analyzed manually with the USCS Genome Browser.

## Generation of stable cell lines

For the generation of stable cell lines, human AMDHD2 isoform 1 was integrated into the FLAG-HA-pcDNA3.1 plasmid (RRID:Addgene_52535) using XbaI and HindIII restriction sites. Cell lines stably overexpressing hAMDHD2 variants were generated by transfection of WT or AMDHD2 KO AN3-12 cells with FLAG-HA-hAMDHD2-pcDNA3.1 plasmids. For each variant, a six-well was transfected with 4 µg of plasmid DNA with Lipofectamine 2000 (Life Technologies) according to the manufacturer's protocol. The selection was performed with 0.4 mg/ml G418 (Gibco) for several weeks.

## Gene editing and genotyping by Sanger sequencing

The specific GFPT1 G451E substitution as well as the KO of GFPT1 and AMDHD2 was engineered in AN3-12 cells (for the AMDHD2 KO also in N2a cells and C2C12 cells) using the CRISPR/Cas9 technology as described previously (*Ran et al., 2013*). DNA template sequences for small guide RNAs were designed online (http://crispor.org, *Supplementary file 1*), purchased from Sigma-Aldrich, and cloned into the Cas9-GFP expressing plasmid PX458 (RRID:Addgene_48138). Corresponding guide and Cas9 expressing plasmids were co-transfected with a single-stranded DNA repair template (Integrated DNA Technologies), using Lipofectamine 3000 (Thermo Fisher Scientific) according to the manufacturer's instructions. For the AN3-12 cells, three different AMDHD2 KO lines were generated, using different guide combinations (clone 1=guide 1 + 2, clone 2=guide 3 + 4, clone 3=guide 1 + 5). GFP-positive cells were singled using FACSAria Fusion sorter and subjected to genotyping. DNA was extracted (DNA extraction solution, Epicentre Biotechnologies) and edited regions were specifically amplified by PCR (primers are listed in *Supplementary file 1*). Sanger sequencing was performed at Eurofins Genomics GmbH (Ebersberg, Germany).

## RNA isolation and qPCR

Cells were collected in QIAzol (QIAGEN) and snap-frozen in liquid nitrogen. Samples were subjected to three freeze/thaw cycles (liquid nitrogen/37°C water bath) before addition of another half of the total QIAzol volume. After incubation for 5 min at RT, 200 µl chloroform were added per 1 ml QIAzol. Samples were vortexed, incubated for 2 min at RT, and centrifuged at 10,000 rpm and 4°C for 15 min. The aqueous phase was mixed with an equal volume of 70% ethanol and transferred to a RNeasy Mini spin column (QIAGEN). The total RNA was isolated using the RNeasy Mini Kit (QIAGEN) and cDNA was subsequently generated by iScript cDNA Synthesis Kit (Bio-Rad). qPCR was performed with Power SYBR Green Master Mix (Applied Biosystems) on a ViiA 7 Real-Time PCR System (Applied Biosystems). GAPDH expression functioned as internal control. All used primers for qPCR analysis are listed in *Supplementary file 1*.

## Anion exchange chromatography mass spectrometry (IC-MS) analysis of UDP-GlcNAc and UDP-GalNAc

Cells were subjected to methanol:acetonitrile:mili-Q ultrapure water (40:40:20 [v:v:v]) extraction. UDP-GlcNAc and UDP-GalNAc (UDP-HexNAc) concentrations were measured using IC-MS analysis. Extracted metabolites were resuspended in 500 µl of Optima LC/MS grade water (Thermo Fisher Scientific) of which 100 µl were transferred to polypropylene autosampler vials (Chromatography Accessories Trott, Germany). The samples were analyzed using a Dionex Ion Chromatography System (ICS5000, Thermo Fisher Scientific) connected to a triple quadrupole MS (Waters, TQ). In brief, 10 µl of the metabolite extract was injected in full loop mode using an overfill factor of 3, onto a Dionex IonPac AS11-HC column (2 mm×250 mm, 4 µm particle size, Thermo Fisher Scientific) equipped with a Dionex IonPac AG11-HC guard column (2 mm×50 mm, 4 µm, Thermo Fisher Scientific). The column temperature was held at 30°C, while the auto sampler was set to 6°C. The metabolite separation was carried using a KOH gradient at a flow rate of 380 µl/min, applying the following gradient conditions: 0–8 min, 30–35 mM KOH; 8–12 min, 35–100 mM KOH; 12–15 min, 100 mM KOH, 15–15.1 min, 10 mM KOH. The column was re-equilibrated at 10 mM for 4 min. UDP-HexNAcs were detected using multiple reaction monitoring mode with the following settings: capillary voltage 2.7 kV, desolvation temperature 550°C, desolvation gas flow 800 L/hr, and collision cell gas flow 0.15 ml/min. The

transitions for UDP-GalNAc, as well as for UDP-GlcNAc were $m/z$ 606 [M-H+]+ for the precursor mass and $m/z$ 385 [M-H+]+ for the first and $m/z$ 282 [M-H+]+ for the second transition mass. The cone voltage was set to 46 V and the collision energy was set to 22 V. UDP-GalNAc eluted at 10.48 min and UDP-GlcNAc eluted at 11.05 min. MS data analysis was performed using the TargetLynx Software (Version 4.1, Waters). Absolute compound concentrations were calculated from response curves of differently diluted authentic standards treated and extracted as the samples.

## Immunoblot analysis

Protein concentration of cell lysates was determined using the Pierce BCA Protein Assay Kit according to the manufacturer's instructions (Thermo Fisher Scientific). Samples were adjusted in 5× LDS sample buffer containing 50 mM DTT. After boiling and a sonication step, equal protein amounts were subjected to SDS-PAGE and blotted on a nitrocellulose membrane using the Trans-Blot Turbo Transfer System (Bio-Rad). All antibodies were used in 5% low-fat milk or 5% BSA in TBS-Tween. After incubation with HRP-conjugated secondary antibody, the blot was developed using ECL solution (Merck Millipore) on a ChemiDoc MP Imaging System (Bio-Rad).

The following antibodies were used in this study: GFPT1 (RRID:AB_10975709, 1:1000), GFPT2 (RRID:AB_2868470, 1:5000), O-Linked N-Acetylglucosamine Antibody (ms, clone RL2, MABS157, Sigma-Aldrich, 1:1000), OGA (RRID:AB_10672079, 1:500), OGT (RRID:AB_2798857, 1:1000), FLAG (RRID:AB_262044, 1:2000), AMDHD2 (ms, S6 clone, in-house produced, 1:500), α-TUBULIN (RRID:AB_477593, 1:5000), rabbit IgG (RRID:AB_2536530,1:5000), and mouse IgG (RRID:AB_2536527, 1:5000).

## Generation of anti-AMDHD2 antibody

To generate monoclonal antibodies directed against AMDHD2, His-tagged human AMDHD2 was expressed in *E. coli*, affinity purified, and used for immunization of 8-week-old male BALB/cJRj mice. The first immunization with 80 µg of recombinant protein was enhanced by Freund's complete adjuvant; subsequent injections used 40 µg protein with Freund's incomplete adjuvant. After multiple immunizations, the serum of the mice was tested for immunoreaction by enzyme-linked immunosorbent assay (ELISA) with the recombinant His-hAMDHD2 protein. In addition, the serum was used to stain immunoblots with lysates of HEK293T cells overexpressing FLAG-HA-hAMDHD2. After this positive testing, cells from the popliteal lymph node were fused with mouse myeloma SP2/0 cells by a standard fusion protocol. Monoclonal hybridoma lines were characterized, expanded, and subcloned according to standard procedures (*Köhler and Milstein, 1975*). Initial screening of clones was performed by ELISA with recombinant His-AMDHD2 protein and immunoblots using FLAG-HA-hAMDHD2 overexpressed in HEK293T cells. Isotyping of selected clones was performed with Pierce Rapid Isotyping Kit (Thermo Fisher Scientific, #26179). Final validation of antibody specificity was done by immunoblots of WT N2a cells compared to cells overexpressing FLAG-HA-hAMDHD2 and AMDHD2 KO cells.

## Expression and purification of human AMDHD2

A pET28a(+)-AMDHD2 plasmid was purchased from BioCat (Heidelberg, Germany), where human AMDHD2 isoform 1 was integrated in pET28a(+) using NdeI and HindIII restriction sites. This vector was used to recombinantly express human AMDHD2 isoform 1 with N-terminal His$_6$tag and a thrombin cleavage site under the control of the T7 promoter in BL21 (DE3) *E. coli*. LB cultures were incubated at 37°C and 180 rpm until an OD$_{600}$ of 0.4–0.6 was reached. Then, protein expression was induced by addition of 0.5 mM isopropyl-β-D-1-thiogalactopyranosid (IPTG) and incubated for 20–22 hr at 20°C and 180 rpm. Before harvest, a sample corresponding to an OD$_{600}$ of 0.5 was taken, lysed in BugBuster Master Mix (Merck Millipore) and the total lysate, the supernatant after centrifugation of the total lysate, as well as the insoluble pellet, which was reconstituted by 8 M urea, were analyzed by SDS-PAGE. The main cultures were harvested and pellets stored at –80°C. The purification buffers were modified from *Bergfeld et al., 2012*. *E. coli* were lysed in 50 mM Tris/HCl pH 7.5, 100 mM NaCl, 20 mM imidazole, 1 mM Tris(2-carboxyethyl)phosphin (TCEP) with complete EDTA-free protease inhibitor cocktail (Roche) and 10 µg/ml DNAseI (Sigma-Aldrich) by sonication. The lysate was clarified by centrifugation and the supernatant loaded on Ni-NTA Superflow affinity resin (QIAGEN). The resin was washed with wash buffer (50 mM Tris-HCl, 100 mM NaCl, 50 mM imidazole, and 1 mM TCEP; pH 7.5) and the protein was eluted with wash buffer containing 250 mM imidazole. The His6-tag was

proteolytically removed using 5 units of thrombin (Sigma-Aldrich) per mg protein overnight at 4°C. AMDHD2 was further purified according to its size on a HiLoad 16/60 Superdex 200 prep grade prepacked column (GE Healthcare) using an ÄKTAprime Chromatography System at 4°C with an SEC buffer containing 50 mM Tris-HCl, 100 mM NaCl, 1 mM TCEP, and 5% glycerol; pH 7.5.

### Site-directed mutagenesis

The AMDHD2 mutations were introduced into the pET28a(+)-AMDHD2 plasmid by site-directed mutagenesis as described previously (*Zheng et al., 2004*; Mutagenesis primers are listed in **Supplementary file 1**). This protocol was also used to integrate an internal His$_6$-tag between Ser300 and Asp301 in human GFPT2 in the plasmid FLAG-HA-hGFPT2-pcDNA3.1 (pcDNA3.1$^{(+)}$, Thermo Fisher Scientific #V79020). This position is equivalent to the internal His$_6$-tag in human GFPT1, which does not interfere with GFPT kinetic properties (*Richez et al., 2007*). The GFPT2 gene with internal His$_6$-tag was subsequently subcloned into the pFL vector for the generation of baculoviruses using XbaI and HindIII entry sites.

### Thermal shift assay

The thermal stability of AMDHD2 was analyzed by thermal shift (ThermoFluor) assays. For this purpose, the proteins were incubated with SYPRO orange dye (Sigma-Aldrich), which binds specifically to hydrophobic amino acids leading to an increased fluorescence at 610 nm when excited with a wavelength of 490 nm. The melting temperature is defined as the midpoint of temperature of the protein-unfolding transition (*Ericsson et al., 2006*). This turning point of the melting curve was extracted from the derivative values of the RFU curve, where a turning point to the right is a minimum. The influence of several divalent cations on the thermal stability of AMDHD2 was tested. For this, the SEC buffer was supplemented with MgCl$_2$, CaCl$_2$, MnCl$_2$, CoCl$_2$, NiCl$_2$, CuSO$_4$, ZnCl$_2$, or CdCl$_2$ at a final concentration of 10 µM. The reaction mixtures were pipetted in white RT-PCR plates and contained 5 µl SYPRO orange dye (1:500 dilution in ddH$_2$O) and 5–10 µg protein in a total volume of 50 µl. The plates were closed with optically clear tape and placed in a Bio-Rad CFX-96 Real-Time PCR machine. The melting curves were measured at 1°C/min at the FRET channel in triplicate measurements and the data were analyzed with CFX Manager (Bio-Rad).

### AMDHD2 activity assay

The deacetylase activity of AMDHD2 was determined by following the cleavage of the amide/peptide bond of the N-acetyl amino sugars GlcNAc6P, GalNAc6P, or GlcNAc6S at 205 nm in UV transparent 96-well microplates (F-bottom, Brand #781614). The assay mix contained 1 mM N-acetyl amino sugar in 50 mM Tris-HCl pH 7.5 and was pre-warmed for 10 min at 37°C in the plate reader. The reaction was started by adding 20 pmol AMDHD2 and was monitored for several minutes at 37°C. The initial reaction rates (0–1 min) were determined by Excel (Microsoft) and the amount of consumed GlcNAc6P was calculated from a GlcNAc6P standard curve. All measurements were performed in duplicates. For the analysis of the impact of several divalent metal ions on the activity of AMDHD2, the protein was incubated for 10 min with 0.1 µM EDTA, and afterward 10 µM divalent was added to potentially restore activity.

### Human AMDHD2 crystallization and crystal soaking

Human AMDHD2 was co-crystallized with a 1.25× ratio (molar) of ZnCl$_2$ at a concentration of 9 mg/ml in sitting drops by vapor diffusion at 20°C. Intergrown crystal plates formed in the PACT *premier* HT-96 (Molecular Dimensions) screen in condition H5 with a reservoir solution containing 0.1 M bis-tris propane pH 8.5, 0.2 M sodium nitrate, and 20% (w/v) PEG3350. In an optimization screen, the concentration of PEG3350 was constant at 20% (w/v), while the pH value of bis-tris propane and the concentration of sodium nitrate were varied. The drops were set up in 1.5 µl protein solution to 1.5 µl precipitant solution and 2 µl protein solution to 1 µl precipitant solution. Best crystals were obtained with a drop ratio of 2 µl protein solution to 1 µl precipitant solution at 0.1 M bis-tris propane pH 8.25, 0.25 M sodium nitrate, and 20% (w/v) PEG3350. 5 mM GlcN6P in reservoir solution was soaked into the crystals for 2–24 hr. For crystal harvesting, the intergrown plates were separated with a needle and 15% glycerol was used as cryoprotectant.

## Data collection and refinement

X-ray diffraction measurements were performed at beamline P13 at PETRA III, DESY, Hamburg (Germany) and beamline X06SA at the Swiss Light Source, Paul Scherrer Institute, Villigen (Switzerland). The diffraction images were processed by XDS (*Kabsch, 2010*). The structure of human AMDHD2 was determined by molecular replacement (*Hoppe, 1957*; *Huber, 1965*) with phenix. phaser (*McCoy, 2007*; *Adams et al., 2010*) using the models of *Bacillus subtilis* AMDHD2 (PDB 2VHL) as search model. The structures were further manually built using COOT (*Emsley et al., 2010*) and iterative refinement rounds were performed using phenix.refine (*Adams et al., 2010*). The structure of GlcN6P soaked crystals was solved by molecular replacement using our human AMDHD2 structure as search model. Geometry restraints for GlcN6P were generated with phenix.elbow software (*Adams et al., 2010*). Structures were visualized using PyMOL (Schrödinger) and 2D ligand-protein interaction diagrams were generated using LigPlot+ (*Laskowski and Swindells, 2011*).

## Analytical SEC

The molecular weight of AMDHD2 and several mutants was determined by analytical SEC on a Superdex 200 Increase 10/300 GL prepacked column (GE Healthcare) using an ÄKTApurifier Chromatography System at 20°C. The measurement was performed with 100µl protein (5 mg/ml) in 50 mM Tris-HCl, 100 mM NaCl, 1 mM TCEP, and 5% glycerol; pH 7.5. All measurements were performed in triplicates and the molecular weight was calculated from a standard curve from proteins with known molecular weights.

## Dynamic Light Scattering

DLS measurements were performed to analyze the size distribution of AMDHD2 in solution. Directly before measurement, 100 µl protein solution was centrifuged for 10 min at 15,000$g$ to remove any particles from solution and 70 µl of the supernatant was transferred into a UV disposable cuvette (UVette 220–1600 nm, Eppendorf #952010051). The cuvette was placed in a Wyatt NanoStar DLS machine and the measurement was performed with 10 frames with 10 s/frame. Data were analyzed with the software Dynamics and converted to particle size distribution functions. The scattering intensity (%) was plotted against the particle radius (nm) in a histogram.

## Baculovirus generation and insect cell expression of GFPT

*Sf21* (RRID:CVCL_0518) suspension cultures were maintained in SFM4Insect HyClone medium with glutamine (GE Lifesciences) in shaker flasks at 27°C and 90 rpm in an orbital shaker. GFPT1 and GFPT2 were expressed in *Sf21* cells using the MultiBac baculovirus expression system (*Berger et al., 2004*). In brief, GFPT (from the pFL vector) was integrated into the baculovirus genome via Tn7 transposition and maintained as bacterial artificial chromosome in DH10EMBacY *E. coli* cells. Recombinant baculoviruses were generated by transfection of *Sf21* with bacmid DNA. The obtained baculoviruses were used to induce protein expression in *Sf21* cells.

## GFPT1 and GFPT2 purification

*Sf21* cells were lysed by sonication in lysis buffer (50 mM Tris/HCl pH 7.5, 200 mM NaCl, 10 mM Imidazole, 2 mM TCEP, 0.5 mM Na$_2$Frc6P, 10% (v/v) glycerol) supplemented with complete EDTA-free protease inhibitor cocktail (Roche) and 10 µg/ml DNAseI (Sigma-Aldrich). Cell debris and protein aggregates were removed by centrifugation and the supernatant was loaded on a Ni-NTA Superflow affinity resin (QIAGEN). The resin was washed with lysis buffer and the protein eluted with lysis buffer containing 200 mM imidazole. The proteins were further purified according to their size on a HiLoad 16/60 Superdex 200 prep grade prepacked column (GE Healthcare) using an ÄKTAprime Chromatography System at 4°C with an SEC buffer containing 50 mM Tris/HCl, pH 7.5, 2 mM TCEP, 0.5 mM Na$_2$Frc6P, and 10% (v/v) glycerol.

## GDH-coupled activity assay and UDP-GlcNAc inhibition

GFPT's amidohydrolysis activity was measured with a coupled enzymatic assay using bovine glutamate dehydrogenase (GDH, Sigma-Aldrich G2626) in 96-well standard microplates (F-bottom, BRAND #781602) as previously described (*Richez et al., 2007*) with small modifications. In brief, the reaction mixtures contained 6 mM Frc6P, 1 mM APAD, 1 mM EDTA, 50 mM KCl, 100 mM potassium-phosphate

buffer pH 7.5, 6.5 U GDH per 96-well and for L-Gln kinetics varying concentrations of L-Gln. For UDP-GlcNAc inhibition assays, the L-Gln concentration was kept at 10 mM. The plate was pre-warmed at 37°C for 10 min and the activity after enzyme addition was monitored continuously at 363 nm in a microplate reader. The amount of formed APADH was calculated with $\varepsilon_{(363\ nm,\ APADH)}$=9100 l*mol$^{-1}$*cm$^{-1}$. Reaction rates were determined by Excel (Microsoft) and $K_m$, $v_{max}$, and $IC_{50}$ were obtained from Michaelis Menten or dose-response curves, which were fitted by Prism 8 software (Graphpad).

## GNA1 expression and purification

The expression plasmid for human GNA1 with N-terminal His$_6$-tag was cloned previously (*Ruegenberg et al., 2020*). Human GNA1 with N-terminal His$_6$-tag was expressed in Rosetta (DE3) *E. coli* cells. LB cultures were incubated at 37°C and 180 rpm until an $OD_{600}$ of 0.4–0.6 was reached. Then, protein expression was induced by addition of 0.5 mM IPTG and incubated for 3 hr at 37°C and 180 rpm. Cultures were harvested and pellets were stored at –80°C. Human GNA1 purification protocol was adopted from *Hurtado-Guerrero et al., 2008* with small modifications. *E. coli* were lysed in 50 mM HEPES/NaOH pH 7.2, 500 mM NaCl, 10 mM imidazole, 2 mM 2-mercaptoethanol, 5% (v/v) glycerol with complete EDTA-free protease inhibitor cocktail (Roche), and 10 µg/ml DNAseI (Sigma-Aldrich) by sonication. The lysate was clarified by centrifugation and the supernatant was loaded on Ni-NTA Superflow affinity resin (Qiagen). The resin was washed with wash buffer (50 mM HEPES/NaOH pH 7.2, 500 mM NaCl, 50 mM imidazole, and 5% (v/v) glycerol) and the protein was eluted with wash buffer containing 250 mM imidazole. Eluted protein was then dialyzed against storage buffer (20 mM HEPES/NaOH pH 7.2, 500 mM NaCl, and 5% (v/v) glycerol).

## GNA1 and GNA1-coupled activity assays

The activity of human GNA1 was measured in 96-well standard microplates (F-bottom, BRAND #781602) as described previously (*Li et al., 2007*). For kinetic measurements, the assay mixture contained 0.5 mM Ac-CoA, 0.5 mM DTNB, 1 mM EDTA, 50 mM Tris/HCl pH 7.5, and varying concentrations of D-GlcN6P. The plates were pre-warmed at 37°C and reactions were initiated by addition of GNA1. The absorbance at 412 nm was followed continuously at 37°C in a microplate reader. The amount of produced TNB, which matches CoA production, was calculated with $\varepsilon_{(412\ nm,\ TNB)}$=13,800 l*mol$^{-1}$*cm$^{-1}$. Typically, GNA1 preparations showed a $K_m$ of 0.2±0.1 mM and a $k_{cat}$ of 41±8 s$^{-1}$.

GFPT's D-GlcN6P production was measured in a GNA1-coupled activity assay following the consumption of AcCoA at 230 nm in UV transparent 96-well microplates (F-bottom, Brand #781614) as described by *Li et al., 2007*. In brief, the assay mixture contained 10 mM L-Gln, 0.1 mM AcCoA, 50 mM Tris/HCl pH 7.5, 2 µg hGNA1, and varying concentrations of Frc6P. The plates were incubated at 37°C for 4 min and reactions started by adding L-Gln. Activity was monitored continuously at 230 nm and 37°C in a microplate reader. The amount of consumed AcCoA was calculated with $\varepsilon_{(230\ nm,\ AcCoA)}$=6436 l*mol$^{-1}$*cm$^{-1}$. As UDP-GlcNAc absorbs light at 230 nm, the GNA-1-coupled assay cannot be used to analyze UDP-GlcNAc effects on activity.

## CRISPR/Cas9-mediated generation of transgenic mice

CRIPSR/Cas9-mediated generation of AMDHD2 KO mice was performed by ribonucleoprotein complex injection in mouse zygotes. Guide RNAs (crRNAs) targeting exon 4 of the *Amdhd2* locus were designed online (http://crispor.tefor.net/) and purchased from IDT. crRNA and tracrRNA were resuspended in injection buffer (1 mM Tris-HCl pH 7.5, 0.1 mM EDTA) and annealed at 1:1 molar concentration in a thermocycler (95°C for 5 min, ramp down to 25°C at 5°C/min). To prepare the injection mix (100 µl), two guide RNAs and the Cas9 enzyme (*Streptococcus pyogenes*, NEB) were diluted to a final concentration of 20 ng/µl each in injection buffer. The mix was incubated for 10–15 min at room temperature to allow ribonucleoprotein complex assembly. After centrifugation, 80 µl of the supernatant was passed through a filter (Millipore, UFC30VV25). Both centrifugation steps were performed for 5 min at 13,000 rpm at room temperature. The filtered injection mix was used for zygote injections.

## Mouse zygote microinjections

After 48 hr, 3- to 4-week-old C57Bl/6J females were superovulated by intraperitoneal injection of Pregnant Mare Serum Gonadotropin (5 IU) followed by intraperitoneal injection of Human Chorionic

Gonadotropin hormone (5 IU Intervet Germany). Superovulated females were mated with 10- to 20-week-old stud males. The mated females were euthanized the next day and zygotes were collected in M2 media (Sigma-Aldrich) supplemented with hyaluronidase (Sigma-Aldrich).

Fertilized oocytes were injected into the pronuclei or cytoplasma with the prepared CRIPSR/Cas9 reagents. Injections were performed under an inverted microscope (Zeiss Axio Observer) associated micromanipulator (Eppendorf NK2) and the microinjection apparatus (Eppendorf Femtojet) with in-house pulled glass capillaries. Injected zygotes were incubated at 37°C, 5% $CO_2$ in KSOM (Merck Millipore) until transplantation. Twenty-five zygotes were surgically transferred into one oviduct of pseudo-pregnant CD1 female mice.

All procedures have been performed in our specialized facility, followed all relevant animal welfare guidelines and regulations, and were approved by LANUV NRW 84-02.04.2015.A025.

## Isolation of mouse genomic DNA from ear clips

Ear clips were taken by the Comparative Biology Facility at the Max Planck Institute for Biology of Ageing (Cologne, Germany) at weaning age (3–4 weeks of age) and stored at –20°C until use. 150 µl $ddH_2O$ and 150 µl direct PCR Tail Lysis reagent (Peqlab) were mixed with 3 µl proteinase K (20 mg/ml in 25 mM Tris-HCl, 5 mM $Ca_2Cl$, pH 8.0, Sigma-Aldrich). This mixture was applied to the ear clips, which were then incubated at 56°C overnight (maximum 16 hr) shaking at 300 rpm. Proteinase K was inactivated at 85°C for 45 min without shaking. The lysis reaction (2 µl) was used for genotyping PCR without further processing. For genotyping of mouse genomic DNA DreamTaq DNA polymerase (Thermo Fisher Scientific) was used.

## Alignments

Following UnitProt IDs were used for the protein sequence alignment of AMDHD2: *Homo sapiens* isoform 1: Q9Y303-1, *Mus musculus*: Q8JZV7, *C. elegans*: P34480, *Candida albicans*: Q9C0N5, *E. coli*: P0AF18, and *B. subtilis*: O34450. ClustalOmega (https://www.ebi.ac.uk/Tools/msa/clustalo/) was used to generate a multiple sequence alignment (*Sievers et al., 2011*). The alignment was formatted with the ESPript3 server (espript.ibcp.fr/) (*Robert and Gouet, 2014*) and further modified.

## Statistical analysis

Data are presented as mean ± SEM/SD or as mean + SEM/SD. The mean of technical replicates is plotted for each biological replicate. Biological replicates represent different passages of the cells that were seeded on independent days. Statistical significance was calculated using GraphPad Prism (GraphPad Software, San Diego, CA). The statistical test used is indicated in the respective figure legend. Significance levels are *p<0.05, **p<0.01, ***p<0.001 versus the respective control.

## Acknowledgements

The authors thank all MSD and UB laboratory members as well as L Kurian, H Bazzi, and D Vilchez for helpful discussions. The FLAG-HA-hGFPT-2-pcDNA3.1 plasmid was kindly provided by C Geisen (Max Planck Institute for Biology of Ageing, MPI-AGE). The authors thank Y Hinze, S Perin, and P Giavalisco from the MPI-AGE metabolomics core facility. The authors thank K Folz-Donahue, L Schumacher, A Just, and C Kukat from the MPI-AGE FACS and imaging core facility. The authors thank F Metge, and J Boucas from the MPI-AGE bioinformatics core facility. The authors thank I Vogt from the MPI-AGE transgenesis core facility. The authors thank the MPI-AGE comparative biology facility. The authors thank M Grzonka for support with the E14 mESCs. The authors are grateful to S Birkmann for support in the insect cell maintenance. The authors thank I Grimm and S Schäfer for their help with the AMDHD2 production and D Feind for the site-directed mutagenesis of monomeric AMDHD2 mutants. Crystals were grown in the Cologne Crystallization facility (C2f). The authors thank the staff of beamline X06SA at the Swiss Light Source, Paul Scherrer Institute, Villigen (Switzerland) and beamline P13 at PETRA III, DESY, Hamburg (Germany) for their support during data collection. This work was supported by the German Federal Ministry of Education and Research (BMBF, Grant 01GQ1423A EndoProtect), by the German Research Foundation (DFG, Projektnummer 73111208-SFB 829, B11 and B14), by the European Commission (ERC-2014-StG-640254-MetAGEn), and by the Max Planck Society. The Cologne Crystallization Facility C2f was supported by DFG Grant INST 216/949-1 FUGG.

## Additional information

### Funding

| Funder | Grant reference number | Author |
| --- | --- | --- |
| Bundesministerium für Bildung und Forschung | 01GQ1423A EndoProtect | Sabine Ruegenberg Stephan Miethe Martin Sebastian Denzel |
| Deutsche Forschungsgemeinschaft | 73111208-SFB 829 | Ulrich Baumann Martin Sebastian Denzel |
| H2020 European Research Council | ERC-2014-StG-640254-MetAGEn | Martin Sebastian Denzel |
| Max Planck Institute for Biology of Ageing | Open Access Funding | Virginia Kroef |
| Deutsche Forschungsgemeinschaft | SCHE1562/8-1 | Bernhard Schermer |
| Deutsche Forschungsgemeinschaft | SFB1403, 414786233, A09 | Bernhard Schermer |
| Deutsche Forschungsgemeinschaft | DE 2326/3-1 | Martin Sebastian Denzel |

The funders had no role in study design, data collection and interpretation, or the decision to submit the work for publication.

### Author contributions

Virginia Kroef, Sabine Ruegenberg, Conceptualization, Data curation, Formal analysis, Investigation, Methodology, Validation, Visualization, Writing - original draft, Writing – review and editing; Moritz Horn, Conceptualization, Data curation, Formal analysis, Methodology; Kira Allmeroth, Data curation, Formal analysis; Lena Ebert, Ulrich Elling, Data curation, Investigation, Methodology; Seyma Bozkus, Stephan Miethe, Investigation; Bernhard Schermer, Investigation, Methodology, Project administration, Resources; Ulrich Baumann, Methodology, Resources, Supervision, Writing – review and editing; Martin Sebastian Denzel, Conceptualization, Formal analysis, Funding acquisition, Investigation, Methodology, Project administration, Resources, Supervision, Writing - original draft, Writing – review and editing

### Author ORCIDs

Virginia Kroef http://orcid.org/0000-0003-3695-911X
Sabine Ruegenberg http://orcid.org/0000-0001-5292-9610
Kira Allmeroth http://orcid.org/0000-0002-2659-6776
Martin Sebastian Denzel http://orcid.org/0000-0002-5691-3349

### Ethics

All procedures have been performed in our specialized facility, followed all relevant animal welfare guidelines and regulations, and were approved by LANUV NRW 84-02.04.2015.A025.

### Decision letter and Author response

Decision letter https://doi.org/10.7554/eLife.69223.sa1
Author response https://doi.org/10.7554/eLife.69223.sa2

## Additional files

### Supplementary files
• Transparent reporting form
• Supplementary file 1. Primers used in this study.

## Data availability

Structural data reported in this study have been deposited in the Protein Data Bank with the accession codes 7NUT [https://doi.org/10.2210/pdb7NUT/pdb] and 7NUU [https://doi.org/10.2210/pdb7NUU/pdb].

The following datasets were generated:

| Author(s) | Year | Dataset title | Dataset URL | Database and Identifier |
|---|---|---|---|---|
| Ruegenberg S, Kroef V, Baumann U, Denzel MS | 2021 | Crystal structure of human AMDHD2 in complex with Zn and GlcN6P | https://www.wwpdb.org/pdb?id=pdb_00007nut | Protein Data Bank, PDB-7NUT |
| Ruegenberg S, Kroef V, Baumann U, Denzel MS | 2021 | Crystal structure of human AMDHD2 in complex with Zn | https://www.rcsb.org/structure/7NUU | RCSB Protein Data Bank, PDB-7NUU |

The following previously published datasets were used:

| Author(s) | Year | Dataset title | Dataset URL | Database and Identifier |
|---|---|---|---|---|
| Kurian L, Frank S | 2019 | yylncT acts as a gatekeeper of the mesodermal transcriptional program by local modulation of DNMT3B [human_1] | https://www.ncbi.nlm.nih.gov/geo/query/acc.cgi?acc=GSE115575 | NCBI Gene Expression Omnibus, GSE115575 |
| Vilchez D | 2017 | Somatic increase of CCT8 mimics proteostasis of human pluripotent stem cells and extends *C. elegans* lifespan | https://www.omicsdi.org/dataset/pride/PXD005123 | PRIDE, PXD005123 |

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
