## [Editor Report]

This manuscript describes an interesting regulation of the hexosamine biosynthetic pathway (HBP) that is relative specific to the mouse embryonic stem cells (mESC). HBP produces UDP-N-acetylglucosamine, which is used in various protein glycosylation events, thus regulating many biological pathways. Understanding this pathway and its regulation is thus of fundamental significance.

---

## [Decision Letter]

**Decision letter after peer review:**

Thank you for submitting your article "GFAT2 and AMDHD2 act in tandem to control the hexosamine biosynthetic pathway" for consideration by *eLife*. Your article has been reviewed by 2 peer reviewers, including Hening Lin as Reviewing Editor and Reviewer #1, and the evaluation has been overseen by Michael Marletta as the Senior Editor.

Essential revisions:

1) Please carry out the rescue experiments recommended by Reviewer 2.

2) Make revisions according to Reviewer 2's comments on statistics (such as number of replicates).

*Reviewer #1 (Recommendations for the authors):*

I really liked the manuscript and thought the conclusions are well supported by the data.

*Reviewer #2 (Recommendations for the authors):*

Introduction:

The introduction, while informative, is "scattergun". In other words, it addresses multiple different aspects, e.g. the HBP, UDP-GlcNAc-dependent glycosylation events, previous work leading up to the publications, without getting to the core of what the paper is about. Both GFAT1 and GFAT2 should be discussed, including known differential expression data on the latter. A clear rationale for the screen should be given. The section describing the steps of the HBP (second paragraph) is missing a lot of references, which must be added.

Figure 1 C:

The authors need to show the full western blot for their new AMDHD2 antibody – not just the AMDHD2 band. Otherwise, the reader is unable to assess the amount of non-specific binding by the antibody. Furthermore, the authors should test this antibody in cell lines derived from multiple species (e.g. Rat PC-12, Mouse N2A, Human SH SY5Y) so the reader is aware of the potential for this antibody to react with AMDHD2 homologs from different species. Can the antibody be used for IP or IF? Has this been tested?

Figure 1 D:

The authors should include a rescue experiment, where AMDHD2 KO's are transfected with AMDHD2. If the number of live cells is subsequently reduced to control levels, then it is acceptable to conclude that the increase in cell survival is due to the AMDHD2 KO, and not non-specific mutagenesis of another gene by CRISPR-Cas9.

Figure 1 G:

There are two controls that should be added here to confirm that AMDHD2 KO is the sole reason for TM resistance (and the up-regulated UDP-GlcNAc levels). Firstly, as for Figure 1 D, the authors need to transfect their AMDHD2 KOs with AMDHD2 and measure UDP-GlcNAc levels, to confirm that it is the KO of AMDHD2 that is causing the up-regulated UDP-GlcNAc levels. Secondly, UDP-GlcNAc levels in the TM resistant clones should be compared to the AMDHD2 KO. If UDP-GlcNAc levels in the TM resistant clones are higher, this contradicts the authors' assertion that TM-resistance is solely due to AMDHD2 LOF.

Figure 1 —figure supplement 1:

In part (a), the authors only assess cell viability in duplicate. Plotting error bars from duplicate measurements is not statistically sound, and this experiment should be done at least in triplicate, and given these are fairly routine assays having 6 independent measurements would be desirable. As for Figure 1 G, the authors need to confirm cell survival and UDP-GlcNAc levels in the same experiment, and plot the associated data as a single plot with the associated statistics.

Figure 2 B:

The authors need to show statistical significance on their graphs. In the text, the authors state that NiCl2 and CoCl2 "restored and even increased the AMDHD2 activity". The authors need to show whether the differences between control and EDTA + NiCl2/ CoCl2 are statistically significant, and display this on the graph.

A comment of the AMDHD2 heterozygote KO mice:

The authors state that "no obvious phenotype" was observed. In the interests of transparency, the authors should provide the negative data and associated behavioural assays and/or anatomical measurements used to arrive at this conclusion.

Figure 2:

The authors state that AMDHD2 is an "obligate dimer", and there are two points of feedback here.

1) The term "obligate dimer" specifically refers to proteins that must be dimers in order to either remain soluble or catalytically active. The authors report that AMDHD2 monomers form during SEC, indicating AMDHD2 folding is not dependent on dimerization.

2) The authors do not provide in vitro data showing the monomer is catalytically inactive – this is inferred from the crystal structure. Can the authors express a rational mutation of AMDHD2, where the dimer cannot form, and display its' catalytic inactivity in vitro? It could alternatively be hypothesised that AMDHD2 monomers display reduced, but not abolished, catalytic activity.

An alternative here might be to tone down the conclusions and suggest that these data are compatible with it being an obligate dimer.

Figure 3 A:

The authors should provide Coomassie stained gels including the pellet fractions for each constructs' expression, to clearly show accumulation of insoluble AMDHD2 mutations in the insoluble fraction.

Also, have the authors performed thermal shift assays (e.g. thermofluor) for the I38T and G256R mutants? These mutations are present in the soluble fraction following induction, and reduced stability in a thermofluor assay would support their assertion that protein folding is impaired.

Just because a missense variant of an enzyme cannot be expressed, does not mean that the mutation causes protein unfolding/ insolubility. Frequently, even catalytically inactive (rational, structural guided) mutants of enzymes prove harder to express and solubilise than their WT equivalents, even though the mutation itself does not impair protein folding. Have the authors tried any additional experiments to confirm the mutations reduce protein stability/ solubility?

Figure 4 A:

Why do the authors only use n = 2? This is statistically improper. Indeed, throughout the figure the n values are inconsistent (for example, in 4 H, n = 4 for a similar RT-qPCR experiment). The authors must include an extra biological replicate in Figure 4 A. This applies to all other figures in the paper – the same number of independent observations (and ideally, >= 6) should be used when comparing experiments.

Figure 4 B:

The usage of recombinant hGAFT1 and 2 as standards is not useful here. In both cases, GFAT1 and GFAT2 signals from the cell lines samples are either significantly higher (GFAT2) or lower (GFAT1) than the standards used. This makes any comparison useless, as it is not clear what the concentration : signal relationship is for GFAT_1/2_ during western blotting, and calibration is not possible. This should either be repeated with a range of standards where the GFAT_1/2_ signals lie within the standard range.

There is a mouse monoclonal antibody for GFAT1 commercially available. Why didn't the authors just compare GFAT1 and GFAT2 in the same blot by using the rabbit GFAT2 and mouse GFAT1 antibody followed by multiplexing using secondaries conjugated to different fluorophores?

Figure 4 D:

Was there a specific reason the authors chose to KO AMDHD2 in N2A cells, and not C2C12 or myoblasts? Ideally, the authors should KO AMDHD2 in all the previously used cell lines and measure UDP-GlcNAc levels in all KOs. If UDP-GlcNAc levels are not elevated in C2C12 and myoblast cells (where GFAT1 is predominant), this adds extra strength to their proposed model.

Figure 4 —figure supplement 4:

The authors do not quantify their western blot signals, nor do they carry out their analysis of O-GlcNAc levels in triplicate. Both should be done in light of the fact that protein loading is uneven between different lanes.

Also, E14 mESCs seem to have O-GlcNAc levels that are comparable to N2A and C2C12 cells. This contradicts the authors assertion that O-GlcNAc levels are higher due to reduced feedback inhibition of GFAT2. Importantly, differences in O-GlcNAc levels could be due to different levels of OGT and OGA, and completely independent of the GFAT1/GFAT2 ratio. The authors need to address whether OGT/OGA levels are different in these cell lines, and whether this is the reason why O-GlcNAc levels are different between cell lines.

A comment on the proposed GFAT2 / AMDHD2 model:

To completely convince the reader that AMDHD2 only acts to regulate GFAT2 activity, and not GFAT1 activity, extra experiments are required. Can the authors knock down GFAT2 in mESCs, followed by transfection with GFAT1 in WT and AMDHD2 KO cells? If UDP-GlcNAc levels are still elevated in AMDHD2 KO cells, this means that AMDHD2 is not counteracting GFAT_1/2_ activity (as you would expect GFAT1 to be feedback inhibited by UDP-GlcNAc). In WT cells, if you knock-down GFAT2 and transfect with GFAT1, UDP-GlcNAc levels should be lower than control cells (as AMDHD2 is still expressed and active, and GFAT1 is feedback inhibited by UDP-GlcNAc). This experiment would support that AMDHD2 is indeed acting to counter GFAT2 activity in pluripotent stem cells, and that this is only necessary for GFAT2 and not GFAT1.

---

## [Author Response]

Reviewer #2 (Recommendations for the authors):Introduction:The introduction, while informative, is "scattergun". In other words, it addresses multiple different aspects, e.g. the HBP, UDP-GlcNAc-dependent glycosylation events, previous work leading up to the publications, without getting to the core of what the paper is about. Both GFAT1 and GFAT2 should be discussed, including known differential expression data on the latter. A clear rationale for the screen should be given. The section describing the steps of the HBP (second paragraph) is missing a lot of references, which must be added.

The introduction has been modified by adding further information and references for the HBP enzymes. Moreover, we streamlined the introduction to better focus it on the aim of our screen.

Figure 1 C:The authors need to show the full western blot for their new AMDHD2 antibody – not just the AMDHD2 band. Otherwise, the reader is unable to assess the amount of non-specific binding by the antibody. Furthermore, the authors should test this antibody in cell lines derived from multiple species (e.g. Rat PC-12, Mouse N2A, Human SH SY5Y) so the reader is aware of the potential for this antibody to react with AMDHD2 homologs from different species. Can the antibody be used for IP or IF? Has this been tested?

We are happy to provide additional data on the AMDHD2 antibody. We provide a full Western blot in the source data, which confirms the specificity of the antibody: we detect only one specific band in four distinct murine cell lines (AN3-12 mESC, E14 mESCs, N2a cells, C2C12 cells)(Figure 5-source data 10-11). Moreover, we tested the AMDHD2 antibody in a human cell line (H9 hESCs), and detect an additional band, which is in accordance with a human-specific AMDHD2 isoform of about 63 kDa (https://www.uniprot.org/uniprot/Q9Y303).

In addition, we performed a pulldown experiment with the newly generated antibody in WT and AMDHD2 KO mESCs to corroborate its specificity (see Author response image 1). We compared beads coupled to the AMDHD2 antibody (+AB) to beads without coupling to the AB (-AB) to exclude unspecific binding of proteins to the beads. Indeed, we were able to enrich for AMDHD2 protein in the eluate of WT cells but not of AMDHD2 KO cells, which serves as a negative control, confirming the high specificity of the AMDHD2 antibody.

**Author response image 1. sa2fig1:** 

Figure 1 D:The authors should include a rescue experiment, where AMDHD2 KO's are transfected with AMDHD2. If the number of live cells is subsequently reduced to control levels, then it is acceptable to conclude that the increase in cell survival is due to the AMDHD2 KO, and not non-specific mutagenesis of another gene by CRISPR-Cas9.

We appreciate the reviewer’s suggestion of performing rescue experiments to confirm the dependence of the observed phenotype on AMDHD2 disruption. In order to address this point, we generated mESC lines stably transfected with FLAG-HA-hAMDHD2 in the WT and in the AMDHD2 KO background. As a control, in parallel we generated cells stably expressing FLAG-HA-hAMDHD2 D294A. In bacteria, this is an inactive mutant^3^ and we show for the first time that it also abolishes activity in human AMDHD2. Loss of activity was confirmed in Figure 4C. In line with our model, expression of mutant AMDHD2 D294A did not alter cell viability of AMDHD2 disrupted cells. In contrast, overexpression of WT FLAG-HA-hAMDHD2 indeed results in a significant rescue in the AMDHD2 KO line, decreasing cell viability in a TM resistance assay, while no effect was observed in WT cells (Figure 2D). In accordance, we observed a similar pattern when we measured UDP-GlcNAc levels (Figure 2C). The missing full rescue to WT levels despite the high rate of overexpression could be caused by differences of human vs. mouse AMDHD2 (although 90% conservation, see alignment Figure 3—figure supplement 8). Additionally, we cannot exclude that the N-terminal tag might disturb enzyme functionality in vivo.

However, we can exclude non-specific effects by the mutagenesis of other loci during CRISPR-Cas9 gene editing, since we analyzed 3 independent AMDHD2 KO lines that were generated using different combinations of target guides. All show similar tunicamycin resistance and UDP-GlcNAc levels (Figure 1E, Figure 1—figure supplement 1D, Figure 2B, Figure 2—figure supplement 1).

Together the new rescue data as well as the use of three independent AMDHD2 KO lines, generated with different guide combinations, with a similar phenotype, confirm the phenotype’s dependence on AMDHD2.

Figure 1 G:There are two controls that should be added here to confirm that AMDHD2 KO is the sole reason for TM resistance (and the up-regulated UDP-GlcNAc levels). Firstly, as for Figure 1 D, the authors need to transfect their AMDHD2 KOs with AMDHD2 and measure UDP-GlcNAc levels, to confirm that it is the KO of AMDHD2 that is causing the up-regulated UDP-GlcNAc levels. Secondly, UDP-GlcNAc levels in the TM resistant clones should be compared to the AMDHD2 KO. If UDP-GlcNAc levels in the TM resistant clones are higher, this contradicts the authors' assertion that TM-resistance is solely due to AMDHD2 LOF.

We have now added substantial data to our paper that address the reviewer’s concerns.

To address the first point, we performed the recommended rescue experiment and observed indeed a reduction of UDP-GlcNAc levels, indicating a dependence of HBP flux on AMDHD2 expression levels (see discussion above).

To address the second point, we have also added new experimental data:

First, in working with the AMDHD2 KO cells generated by various means (ENU mutagenesis or by gene targeting) we consistently observed a similar level of TM resistance, that was around 4-fold increased compared to WT cells (Figure 1E, Figure 1—figure supplement 1A,D). Unfortunately, the initial clones from our ENU screen were not available to perform further experiments. However, it is worth pointing out that we generally avoid performing additional follow up work in the original ENU-generated cells, as these carry up to 100 additional SNPs that might affect protein function. After the first identification of AMDHD2 as a candidate by sequencing, it was therefore much better to target the locus specifically by CRISPR/Cas9 gene editing and then do follow up work in this clean cell line. It is these cells that establish a role of AMDHD2 in the HBP of mESCs.

Second, to further corroborate our data we performed an additional insertional mutagenesis screen. In this screen, that is based on a gene trapping method, 20% of all tested TM resistant clones showed an insertion in AMDHD2. Four independent clones derived from this screen showed similar TM resistance and UDP-HexNAc levels as CRISPR/Cas9 generated AMDHD2 KO cells (Figure 2B, Figure 2—figure supplement 1).

Together, these data provide evidence that AMDHD2 is the sole cause for elevated HBP flux, mediating TM resistance.

Figure 1 —figure supplement 1:In part (a), the authors only assess cell viability in duplicate. Plotting error bars from duplicate measurements is not statistically sound, and this experiment should be done at least in triplicate, and given these are fairly routine assays having 6 independent measurements would be desirable. As for Figure 1 G, the authors need to confirm cell survival and UDP-GlcNAc levels in the same experiment, and plot the associated data as a single plot with the associated statistics.

As discussed in the comment for Figure 1G, the initial clones from our ENU screen were not available to perform further replicates. To avoid confusion, we removed the error bars and only display the data of one representative experiment (Figure1—figure supplement 1A).

Figure 2 B:The authors need to show statistical significance on their graphs. In the text, the authors state that NiCl2 and CoCl2 "restored and even increased the AMDHD2 activity". The authors need to show whether the differences between control and EDTA + NiCl2/ CoCl2 are statistically significant, and display this on the graph.

Thanks for pointing out the missing statistics. We implemented these and rephrased our manuscript to “Both CoCl2 and ZnCl2 restored and ZnCl2 even increased the AMDHD2 activity (Figure 3B).”.

A comment of the AMDHD2 heterozygote KO mice:The authors state that "no obvious phenotype" was observed. In the interests of transparency, the authors should provide the negative data and associated behavioural assays and/or anatomical measurements used to arrive at this conclusion.

Indeed, we only did macroscopic analysis of heterozygous mice to an age up to 1 year. As of now, we did not perform further experiments such as behavioral, anatomical, histological or molecular analyses. A detailed analysis of a conditional AMDHD2 KO mouse is planned for future experiments but not focus of this study. To be transparent with our data, we rephrased the manuscript to “Heterozygous animals however, did not show any macroscopic changes, although further analysis is still missing and alterations on a behavioral, anatomical, histological or molecular level cannot be excluded.”

Figure 2:The authors state that AMDHD2 is an "obligate dimer", and there are two points of feedback here.1) The term "obligate dimer" specifically refers to proteins that must be dimers in order to either remain soluble or catalytically active. The authors report that AMDHD2 monomers form during SEC, indicating AMDHD2 folding is not dependent on dimerization.2) The authors do not provide in vitro data showing the monomer is catalytically inactive – this is inferred from the crystal structure. Can the authors express a rational mutation of AMDHD2, where the dimer cannot form, and display its' catalytic inactivity in vitro? It could alternatively be hypothesised that AMDHD2 monomers display reduced, but not abolished, catalytic activity.An alternative here might be to tone down the conclusions and suggest that these data are compatible with it being an obligate dimer.

We thank the reviewer for this nice suggestion! As part of the revision process, we generated three mutants to further analyze the role of dimerization for AMDHD2’s catalytic activity and characterized them in analytical size-exclusion chromatography measurements and activity assays (new Figures 3 F-I). Ile280 is located at the dimer interface and we introduced Glu and Arg side chains to disrupt dimerization (Figure 3F). We confirmed by analytical size-exclusion chromatography that the dimer cannot form in the I280E and I280R mutants (Figure 3 G,H). Indeed, the mutants I280E and I280R completely lose catalytic activity in activity assays (Figure 3I) showing that AMDHD2 forms an obligate dimer. Two side chains, His242 and Arg243, are of particular interest in this dimerization, because they mediate interactions from one monomer to the active site at the other monomer and are involved in the phosphate binding of the sugar (Figure 3E). Therefore, we generated a double mutant H242A/R243A, whose dimerization seemed not to be disturbed according to analytical size-exclusion chromatography measurements (Figure 3G,H). Interestingly, also this mutant showed no activity in our in vitro assay, indicating an essential role of those two residues for the catalytic activity of AMDHD2.

In the first version of the manuscript, we included a representative chromatogram of a preparative size-exclusion chromatography from the wildtype protein, which corresponded to a monomeric AMDHD2 and showed a dynamic light scattering measurement, which corresponded to a dimeric protein. Based on this, we speculated that the dimer might not be very stable. Because the new analytical size-exclusion chromatography measurements were much more accurate than our previous preparative size-exclusion chromatography, we decided to exclude the previous SEC chromatogram (previous Figure 2—figure supplement 3A) and re-phrased our manuscript accordingly.

Figure 3 A:The authors should provide Coomassie stained gels including the pellet fractions for each constructs' expression, to clearly show accumulation of insoluble AMDHD2 mutations in the insoluble fraction.

Prompted by the reviewers comment we repeated the test expressions for the AMDHD2 loss-of-function mutants, which showed a reduced solubility in the first version of the manuscript and included now the new Figure 4 —figure supplement 1, showing Coomassie stained gels including the pellet fractions of insoluble AMDHD2. AMDHD2 G265R, which was previously observed to be partially soluble was more difficult to express during the revision experiments. However, we stated before “Also, the I38T and G265R substitutions reduced soluble expression, indicating disturbed protein folding” and our new data support this conclusion.

Also, have the authors performed thermal shift assays (e.g. thermofluor) for the I38T and G256R mutants? These mutations are present in the soluble fraction following induction, and reduced stability in a thermofluor assay would support their assertion that protein folding is impaired.Just because a missense variant of an enzyme cannot be expressed, does not mean that the mutation causes protein unfolding/ insolubility. Frequently, even catalytically inactive (rational, structural guided) mutants of enzymes prove harder to express and solubilise than their WT equivalents, even though the mutation itself does not impair protein folding. Have the authors tried any additional experiments to confirm the mutations reduce protein stability/ solubility?

We fully agree with the reviewer that additional data supporting a reduced stability of our mutants would strengthen our manuscript. We now performed these thermal shift assays (see Author response image 2) and find a reduced stability for the catalytic inactive mutants T185A and G265R. A destabilization in the T185A could be explained by its close proximity to the Zn2+ co-factor, which stabilizes AMDHD2. The replacement of Gly by Arg in G265R might be incompatible with the tertiary structure. The stability of I38T and the known inactive D294A mutant was not affected in these thermal shift assays.

However, we were concerned about the informative value from these measurements, because the thermal shift assay most likely reflects the unfolding of the bigger deacetylase domain of AMDHD2. I38T is located in the DUF domain, which is smaller as the deacetylase domain and formed by residues of the N- and C-terminus (Figure 3—figure supplement 8). Therefore, the unfolding of the DUF domain might be dependent on the unfolding of the deacetylase domain or the DUF domain unfolding might be masked by the fluorescence signal of the bigger deacetylase domain. Given these concerns, we decided to exclude the thermal shift assays from the manuscript.

Figure 4 A:Why do the authors only use n = 2? This is statistically improper. Indeed, throughout the figure the n values are inconsistent (for example, in 4 H, n = 4 for a similar RT-qPCR experiment). The authors must include an extra biological replicate in Figure 4 A. This applies to all other figures in the paper – the same number of independent observations (and ideally, >= 6) should be used when comparing experiments.

We agree with the reviewer and increased the number of replicates accordingly. We replaced the graph with new measurements repeated in a set of n=4 (Figure 5A).

Figure 4 B:The usage of recombinant hGAFT1 and 2 as standards is not useful here. In both cases, GFAT1 and GFAT2 signals from the cell lines samples are either significantly higher (GFAT2) or lower (GFAT1) than the standards used. This makes any comparison useless, as it is not clear what the concentration : signal relationship is for GFAT_1/2_ during western blotting, and calibration is not possible. This should either be repeated with a range of standards where the GFAT_1/2_ signals lie within the standard range.There is a mouse monoclonal antibody for GFAT1 commercially available. Why didn't the authors just compare GFAT1 and GFAT2 in the same blot by using the rabbit GFAT2 and mouse GFAT1 antibody followed by multiplexing using secondaries conjugated to different fluorophores?

Indeed, the protein concentrations of GFPT_1/2_ of our cell lysates are not within the range of the used standards of purified human protein. However, we do not use the curve to quantify the absolute amount of protein within our lysates but rather just give an indication of the approximal protein amount. We removed these lanes and only show the band for 10ng of pure hGFPT_1/2_ as a standard (Figure 5B). We highly appreciate the reviewer’s suggestion to detect both GFPT signals on one membrane. However, the use of both antibodies on one membrane is not required, since the comparison between the tested cell lines on each membrane respectively is sufficient to confirm the higher abundance of GFPT2 and lower abundance of GFPT1 in mESCs compared to the other cell lines.

Figure 4 D:Was there a specific reason the authors chose to KO AMDHD2 in N2A cells, and not C2C12 or myoblasts? Ideally, the authors should KO AMDHD2 in all the previously used cell lines and measure UDP-GlcNAc levels in all KOs. If UDP-GlcNAc levels are not elevated in C2C12 and myoblast cells (where GFAT1 is predominant), this adds extra strength to their proposed model.

In our previous work in N2a cells, we confirmed a conserved mechanism of HBP activation by GlcNAc supplementation and by introducing the G451E gain-of-function substitution into GFPT1 that was identified in a previous TM resistance screen in *C. elegans*^4-6^. Based on this knowledge and the already existing GFPT1 gain-of-function cell line, we considered the N2a cell line as a suitable model for further investigation of the HBP regulation.

However, we fully agree with the reviewer’s comment and generated two AMDHD2 KO lines in the C2C12 myoblasts. In accordance with our model, UDP-GlcNAc measurements revealed no significant increase in the AMDHD2 KO cells compared to the WT control (Figure 5—figure supplement 2A-B). These data strengthen the hypothesis that AMDHD2 is not as important for regulation of the HBP in cells relying on GFPT1, as in both cell lines (N2a, C2C12) the deletion of AMDHD2 had no significant effect on UDP-GlcNAc levels.

Figure 4 —figure supplement 4:The authors do not quantify their western blot signals, nor do they carry out their analysis of O-GlcNAc levels in triplicate. Both should be done in light of the fact that protein loading is uneven between different lanes.Also, E14 mESCs seem to have O-GlcNAc levels that are comparable to N2A and C2C12 cells. This contradicts the authors assertion that O-GlcNAc levels are higher due to reduced feedback inhibition of GFAT2. Importantly, differences in O-GlcNAc levels could be due to different levels of OGT and OGA, and completely independent of the GFAT1/GFAT2 ratio. The authors need to address whether OGT/OGA levels are different in these cell lines, and whether this is the reason why O-GlcNAc levels are different between cell lines.

We highly appreciate this suggestion and have performed additional experiments to address them. We provide now the data of four replicates that reveal a significant decrease in the level of O-GlcNAcylated proteins in N2a and C2C12 cells compared to AN3-12 and E14 mESCs (Figure 5G-H). These data are in accordance with the increased levels of UDP-GlcNAc in mESCs, due to the reduced feedback inhibition of the predominant GFPT2 paralog.

Moreover, we thank the reviewer for the insightful comment about the O-GlcNAc regulation by OGT and OGA. To address this point, we measured OGT and OGA levels of the different cell lines via Western blot (Figure 5 I-J). We observed a significant reduction in OGT and OGA protein levels in the more differentiated N2a and C2C12 cells, which interestingly coincides with the reduced O-GlcNAc levels.

As the OGT/OGA ratio is generally thought to determine O-GlcNAcylation levels, we found it noteworthy that the analyzed cells show similar OGT and OGA ratios despite the differences in O-GlcNAc levels. We conclude that indeed the elevated UDP-GlcNAc levels are responsible for O-GlcNAc elevation in mESCs.

A comment on the proposed GFAT2 / AMDHD2 model:To completely convince the reader that AMDHD2 only acts to regulate GFAT2 activity, and not GFAT1 activity, extra experiments are required. Can the authors knock down GFAT2 in mESCs, followed by transfection with GFAT1 in WT and AMDHD2 KO cells? If UDP-GlcNAc levels are still elevated in AMDHD2 KO cells, this means that AMDHD2 is not counteracting GFAT_1/2_ activity (as you would expect GFAT1 to be feedback inhibited by UDP-GlcNAc). In WT cells, if you knock-down GFAT2 and transfect with GFAT1, UDP-GlcNAc levels should be lower than control cells (as AMDHD2 is still expressed and active, and GFAT1 is feedback inhibited by UDP-GlcNAc). This experiment would support that AMDHD2 is indeed acting to counter GFAT2 activity in pluripotent stem cells, and that this is only necessary for GFAT2 and not GFAT1.

In our proposed model AMDHD2 is not only important to regulate GFPT2 activity but is also expressed in cells using GFPT1. Supporting this notion, in cells using GFPT1 the loss of AMDHD2 increases UDP-GlcNAc levels but not significantly (Figure 5D, Figure 5—figure supplement 2B). Moreover, AN3-12 mESCs show a high abundance and constitutive activity of AMDHD2, resulting in a drastic increase of UDP-GlcNAc levels upon loss of AMDHD2.

Thus, in our model, we describe the following scenarios: in GFPT1 cells, a steady UDP-GlcNAc concentration is achieved by strong product feedback inhibition and an AMDHD2-dependent reverse flux. In GFPT2 cells in contrast, the feedback inhibition is much weaker, resulting in elevated UDP-GlcNAc levels and a stronger reliance on the AMDHD2 reaction (Figure 6E). This configuration enabled identification of AMDHD2 in the initial screen.

References

1 Horn, M. et al. Unbiased compound-protein interface mapping and prediction of chemoresistance loci through forward genetics in haploid stem cells. (2018).

2. Bang, M. L. et al. The complete gene sequence of titin, expression of an unusual approximately 700-kDa titin isoform, and its interaction with obscurin identify a novel Z-line to I-band linking system. Circ Res 89, 1065-1072, doi:10.1161/hh2301.100981 (2001).

3. Hall, R. S., Xiang, D. F., Xu, C. and Raushel, F. M. N-Acetyl-D-glucosamine-6-phosphate deacetylase: substrate activation via a single divalent metal ion. Biochemistry 46, 7942-7952, doi:10.1021/bi700543x (2007).

4. Denzel, M. S. et al. Hexosamine pathway metabolites enhance protein quality control and prolong life. Cell 156, 1167-1178, doi:10.1016/j.cell.2014.01.061 (2014).

5. Horn, M. et al. Hexosamine Pathway Activation Improves Protein Homeostasis through the Integrated Stress Response. iScience 23, 100887, doi:10.1016/j.isci.2020.100887 (2020).

6. Ruegenberg, S. et al. Loss of GFAT-1 feedback regulation activates the hexosamine pathway that modulates protein homeostasis. Nat Commun 11, 687, doi:10.1038/s41467-020-14524-5 (2020b).